# *AutoJudger*: An Agent-Driven Framework for Efficient Benchmarking of MLLMs

## Abstract

Evaluating multimodal large language models (MLLMs) is increasingly expensive, as the growing size and cross-modality complexity of benchmarks demand significant scoring efforts. To tackle with this difficulty, we introduce ***AutoJudger***, an agent-driven framework for efficient and adaptive benchmarking of MLLMs that tackles this escalating cost. AutoJudger employs the Item Response Theory (IRT) to estimate the question difficulty and an autonomous evaluation agent to dynamically select the most informative test questions based on the model's real-time performance. Specifically, AutoJudger incorporates two pivotal components: *a semantic-aware retrieval mechanism* to ensure that selected questions cover diverse and challenging scenarios across both vision and language modalities, and *a dynamic memory* that maintains contextual statistics of previously evaluated questions to guide coherent and globally informed question selection throughout the evaluation process. Extensive experiments on four representative multimodal benchmarks demonstrate that our adaptive framework dramatically reduces evaluation expenses, i.e. AutoJudger uses only 4% of the data and 10% computational cost (for evaluating 7B model) to achieve over 90% ranking accuracy with the full-benchmark evaluation results on MMT-Bench.

## 1 Introduction

Motivated by the success of Large Language Models (LLMs) (Achiam et al., 2023; Touvron et al., 2023; Yang et al., 2024; Liu et al., 2024a), Multimodal Large Language Models (MLLMs) (Hurst et al., 2024; Liu et al., 2023; Bai et al., 2025; Chen et al., 2025) have been developed to tackle challenging tasks involving the joint understanding and generation of information across multiple modalities, such as text and images (Li et al., 2024d; 2025). To assess the full spectrum of MLLM capabilities, a growing number of benchmarks have been introduced as illustrated in Figure 1a, spanning diverse domains (Fu et al., 2023; Liu et al., 2024d; Yue et al., 2024; Li et al., 2023; 2024c). However, this also introduces a computational burden for comprehensive evaluation.

Compared to text-only scenarios, the evaluation cost problem becomes more pronounced in multimodal benchmarks, as including visual contexts substantially lengthens the input sequences (Terragni et al., 2024; Xu et al., 2025). In addition, incorporating reasoning-enhancement methods like chain-of-thought (Wei et al., 2022; Guo et al., 2025) and employing ChatGPT to assist in scoring model responses (Liu et al., 2024d; Lu et al., 2023) will further increase the cost. This raises a critical question: **Can we evaluate MLLMs more efficiently without sacrificing reliability?**

To address this problem, a line of studies focuses on exploring efficient benchmarking methods (Perlitz et al., 2023; Polo et al., 2024; Vivek et al., 2023): selecting a subset from the benchmark for efficient evaluation while maintaining consistency with the results obtained from full-set evaluation. Existing approaches are primarily designed for text-only benchmarks, performing stratified sampling based on question categories (Perlitz et al., 2023) and difficulty levels (Zhuang et al., 2023b; Polo et al., 2024) to construct subsets for evaluation. However, multimodal scenarios pose additional challenges: (i) Most multimodal benchmarks do not explicitly assess or characterize question difficulty; (ii) Each image-question pair contains rich multimodal semantic information, merely relying on coarse-grained information like question categories struggles to ensure semantic diversity within the subset; (iii) There exists large performance variance across models. Assigning the same subset of questions to all models may limit the efficiency in distinguishing between models, e.g., evaluating powerful models with too many simple questions provides little information gain.

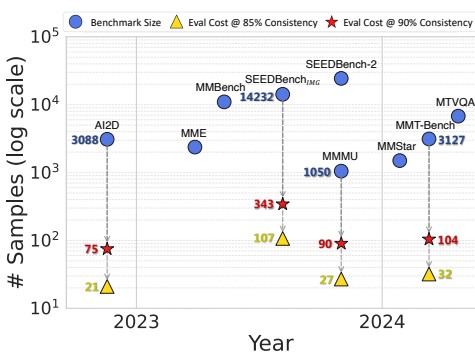
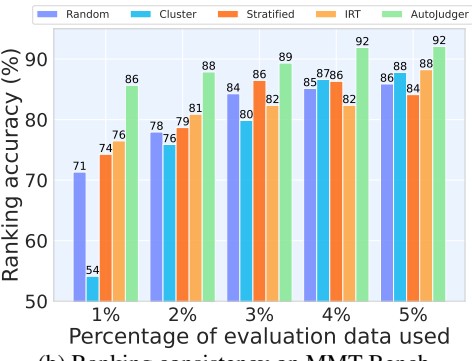

(a) Number of samples in benchmarks over time.     (b) Ranking consistency on MMT-Bench.

Figure 1: **Benchmark scale and efficiency of AutoJudger.** (a) plots the scales of various benchmarks that are commonly adopted in MLLM evaluation. The triangle and pentagon markers indicate the number of samples required by AutoJudger to achieve 85% and 90% consistency with the full-set evaluation results, respectively. (b) compares different efficient benchmarking methods.

Facing above challenges, an ideal evaluation system needs to comprehensively consider factors including the performance of evaluated models, question difficulties and semantics, iteratively constructing subsets during assessment. To tackle such a dynamic decision-making problem, we propose *AutoJudger*, an agent-driven multimodal evaluation framework. We formulate efficient benchmarking as an interview scenario, where an agent powered by MLLM serves as the interviewer, continuously interacting with the environment (the question pool and the evaluated models) to select appropriate questions, dynamically assessing model capabilities throughout the interview.

Furthermore, we design three modules to assist AutoJudger in the interaction with the interview environment and the evaluation process. (i) We collect extensive offline evaluation results of various MLLMs and characterize the difficulty of benchmark questions based on Item Response Theory (IRT) (Cai et al., 2016). This difficulty framework supports subsequent question selection and real-time performance assessment during the interview. (ii) Considering the large scale of the question pool, we augment AutoJudger with a multimodal semantic-aware retrieval module to access the entire benchmark. The retrieval module performs a coarse filtering process, the interviewer agent then conduct fine-grained analysis and select the retrieved candidates. This strategy fully leverages sample-level semantics to ensure richness of the selected subset while maintaining efficiency. (iii) We introduce a dynamic memory module to help the agent summarize about previously tested questions and the model performance. This module assists the agent in making personalized question selections for different models and provides an interpretable analysis of model capabilities.

AutoJudger can be seamlessly integrated into the evaluation of any MLLMs in a plug-and-play manner. We conduct extensive experiments to demonstrate that AutoJudger significantly reduces evaluation costs across different benchmarks while maintaining the reliability and stability. Despite the growing size of modern benchmarks, AutoJudger still achieves remarkable rank consistency—85% and even 90%—with only a small fraction of the samples, as shown in Figure 1. For instance, on MMT-Bench, AutoJudger reaches 92% rank consistency using merely 4% of the data (125 samples).

The key contributions are summarized as follows:

- We propose *AutoJudger*, the first agent-driven framework for efficient benchmarking of MLLMs. Unlike prior static methods, AutoJudger adaptively selects informative questions by interacting with evaluated models, leveraging the reasoning ability of the judging agent to guide the evaluation.

- To jointly capture the question difficulty and the cross-modal semantic diversity, we equip AutoJudger with a semantic-aware retrieval mechanism grounded in Item Response Theory (IRT), ensuring representative question selection. To further enhance adaptivity, we incorporate a dynamic memory that tracks contextual statistics from previously evaluated questions, enabling coherent and globally informed question selection throughout the evaluation process.

- We extensively evaluate 17 MLLMs on four popular multimodal benchmarks with AutoJudger, showing that our adaptive framework significantly reduces evaluation costs — AutoJudger achieves over 90% ranking accuracy on MMT-Bench using only 4% of the data and 10% of the computation.

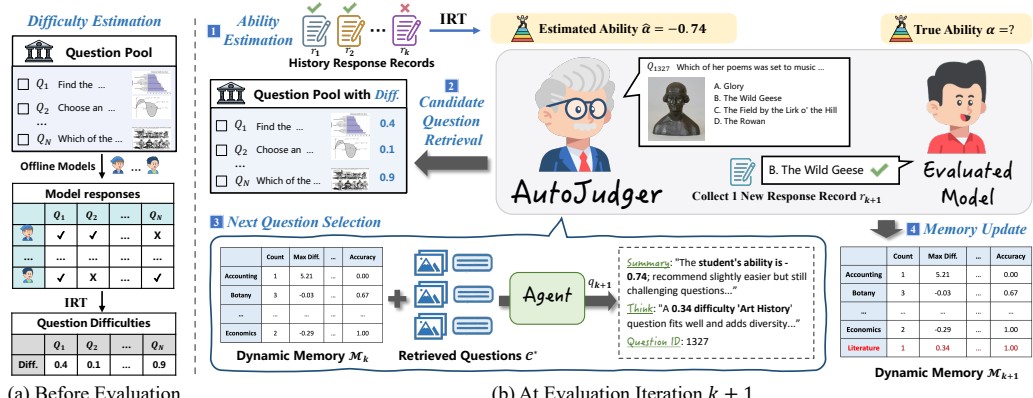

(a) Before Evaluation    (b) At Evaluation Iteration $k+1$

Figure 2: **The framework of AutoJudger.** Before evaluation, the difficulties of question from a benchmark are computed by utilizing a set of offline models. At each evaluation iteration, AutoJudger firstly retrieve the candidate questions based on the estimated ability. Then, AutoJudger selects the most proper question, collect the response from the evaluated model, and update its memory.

## 2 PROBLEM STATEMENT OF EFFICIENT BENCHMARKING

Efficient benchmarking aims to reliably evaluate a model on a specific benchmark with as less expenses as possible. In this work, we constrain the scope of evaluated models to MLLMs, due to the rapidly increasing cost of these models and the lack of prior work in this area. We denote the complete evaluation benchmark as $Q = \{q_i\}_{i=1}^N$ where each $q_i$ is a test question, representing a multimodal question-answer pair. Given a candidate model $m$, the objective is to find a mapping $f : Q \to \hat{Q}$ such that the performance of model $m$ on the selected subset $\hat{Q}$, denoted as $P(m|\hat{Q})$, is consistent with its performance on the full benchmark set $Q$, i.e. $\rho\left(P(m|\hat{Q}), P(m|Q)\right) \geq \sigma$, where $\rho$ represents the consistency scoring function, and $\sigma$ is the consistency threshold. Typically, the consistency is estimated by comparing the ranking of a evaluated model $m_j$ on a group of models $M = \{m_j\}$. Therefore, the problem of efficient benchmarking is formulated as follows:

$$\max_{f:Q\to\hat{Q}} \rho\left(\{P(m_j|\hat{Q})\}_{m_j\in M}, \{P(m_j|Q)\}_{m_j\in M}\right) \quad \text{s.t.} \quad \hat{Q}\subset Q, |\hat{Q}|\leq\delta*|Q| \quad (1)$$

where $\delta$ denotes the compression ratio. In this work, $\delta$ is set to 5% unless otherwise specified.

## 3 *AutoJudger*

To address this problem, we propose ***AutoJudger***, an agent-driven efficient benchmarking framework as shown in Figure 2. Our underlying intuition is that, analogous to human interview, the effective evaluation of a model entails dynamically selecting the most appropriate questions based on its real-time performance. Therefore, we formulate the construction of mapping $f$ as as a dynamic decision-making problem. Specifically, the next question used to evaluate model $m_j$ is selected by considering the memory $\mathcal{M}_k$ of previously attempted questions $Q'_k = \{q_i\}_{i=1}^k$ and responses $\{r_{ij}\}_{i=1}^k$, current model performance $P\left(m_j|\{q_i\}_{i=1}^k\right)$ and the complete evaluation benchmark $Q$:

$$q_{k+1} = f\left(\mathcal{M}_k, P\left(m_j|Q'_k\right), Q\right), \quad \mathcal{M}_k = \{q_i, r_{ij}\}_{i=1}^k \quad (2)$$

To select appropriate samples, AutoJudger should acquire a clear understanding of the difficulty of questions during evaluation. To this end, we first leverage Item Response Theory (IRT) (Cai et al., 2016) to characterize the difficulty $d_i$ of each question $q_i$ before the evaluation (§ 3.1). We then employ an intelligent agent powered by an MLLM to recommend questions of appropriate difficulty. Such recommendation is based on the current estimation of model's ability (§ 3.2). Since the full benchmark $Q$ is too large to be processed efficiently, we design a *semantic-aware retrieval mechanism* to reduce the candidate pool size (§ 3.3). This mechanism also ensures that the selected questions span diverse and challenging scenarios. Subsequently, we prompt an MLLM-based agent to choose the most appropriate question from the retrieved candidate questions (§ 3.4). After the evaluated model answers a newly selected question, we record its response. The history of attempted questions and model responses is summarized by the agent, stored and updated in the *memory* $\mathcal{M}$ of the agent to guide coherent and globally informed question selection during evaluation (§ 3.5).

## 3.1 QUESTION DIFFICULTY ESTIMATION

To preform efficient sample selection, AutoJudger adopts Item Response Theory (IRT) (Cai et al., 2016) to estimate the difficulty of the questions from a specific benchmark by utilizing a set of offline MLLMs before evaluation. Note that, to prevent information leakage, we carefully ensure that there is no overlap between the offline MLLMs and the models under evaluation (see Appendix D).

**Modeling with IRT**   We adopt a logistic IRT model, also known as the Rasch model (Rasch, 1993), which defines the correct probability of a response $r_{ij}$ of model $m_j$ on question $q_i$ as:

$$p\left(r_{ij} \text{ is correct}\right) = \frac{1}{1 + \exp(-(a_j - d_i))} \tag{3}$$

where $a_j$ is the latent ability of the model $m_j$, and $d_i$ is the difficulty of the question $q_i$. Intuitively, a model is more likely to succeed on questions with lower difficulty than its ability level.

**Estimating Question Difficulty**   Given a collection of response records $\{r_{ij}\}$ from a set of previously evaluated offline models $M' = \{m'_i\}$ on benchmark $Q = \{q_i\}$, we fit the Rasch model to estimate the question difficulties $D = \{d_i\}$ via maximum likelihood estimation, using the Bayesian variational framework proposed by (Ding et al., 2024) (please refer to Appendix B for more details). Once estimated, these difficulty scores are fixed and serve as priors for subsequent question selection.

## 3.2 MODEL ABILITY ESTIMATION

Given the estimated question difficulties $D = \{d_i\}$, IRT enables the real-time assessment of model ability based on its response history, supervising the model proficiency during evaluation. When evaluating the model $m_j$ at $k$-th iteration, according to Equation equation 3, we use maximum likelihood estimation to infer the current ability $a_{j,k}$ based on the selected questions $Q'_k$, model responses $\{r_{ij}\}_{i=1}^k$, and the fixed difficulties $D$. We develop a binary search algorithm to efficiently find the optimum for $a_{j,k}$, as detailed in Appendix C. This allows us to track the model's progress over time and guide the adaptive selection of future questions. Notably, we define the model performance $P(m_j|Q'_k)$ as its current ability value $a_{j,k}$ estimated through IRT.

## 3.3 CANDIDATE QUESTION RETRIEVAL

Given the large scale of existing benchmarks, it is impractical for the agent to directly select questions from the entire question pool $Q$. To address this challenge, we design a retrieval strategy to provide the agent with a candidate set $\mathcal{C}_k^*$ with feasible size $|\mathcal{C}_k^*| \ll |Q|$. We aim to select questions that are both appropriate in difficulty and semantically distinct from those previously attempted in $Q'_k$.

$$q_{k+1} = f\left(\mathcal{M}_k, a_j, \mathcal{C}_k^*\right), \quad \mathcal{C}_k^* \subset Q \tag{4}$$

**Initialization**   At the beginning, the ability of the model $P(m_j, \varnothing)$ remains unknown. To build a strong starting point, we adopt a clustering-based strategy to ensure semantic diversity across the initial set $Q'_0$. We first encode each question with a semantic feature extractor (e.g., CLIP (Radford et al., 2021), Qwen2.5-VL (Bai et al., 2025)) to obtain meaningful embeddings. Based on these embeddings, we perform K-means clustering over the entire benchmark to group questions with similar semantics. From each cluster, we uniformly sample a small number of questions to construct the initial candidate pool $Q'_0$. This approach ensures broad semantic coverage and avoids the early-stage selection bias, enabling robust ability estimation and question selection in later adaptive steps.

**During Iteration**   Firstly, we filter out questions that are either too hard or too easy for the current model. We compute the estimated probability $p$ of the target model $m_j$ correctly answering each question $q_i \in Q/Q'_k$, based on its current ability $a_j$ and the question difficulty $d_i$, using the IRT formulation defined in Equation equation 3. We keep a candidate set $\mathcal{C}_k$ to retain only questions whose success probabilities fall within a desirable range (we set $p_{\min}$ as 0.2 and $p_{\max}$ as 0.8):

$$\mathcal{C}_k = \{i \in Q/Q'_k \mid p_{\min} \le p \le p_{\max}\} \tag{5}$$

Subsequently, to encourage semantic diversity, we apply a max-min retrieval strategy. For each candidate $q \in \mathcal{C}_k$, we compute its distance to the previously selected question set $Q'_k$ and select the questions with the maximum distance:

$$\mathcal{C}_k^* = \left\{ q_1^*, ..., q_5^* \mid q^* = \arg\max_{q \in \mathcal{C}_k} \min_{q' \in Q'_k} \text{dist}(q, q') \right\} \tag{6}$$

We retain the top-5 questions that exhibit the greatest Euclidean distance from $Q'_k$ as the final candidate set $\mathcal{C}^*_k$. Consequently, such a semantic-aware retrieval strategy ensures that the selected questions not only align with real-time ability but also introduce novel semantic coverage.

## 3.4 Next Question Selection

Based on the retrieved results $C^*_k$, we prompt the interviewer agent to perform fine-grained analysis on candidates questions and recommend the next question for the tested model as:

$$q_{k+1} = f_\theta(\mathcal{M}_k, a_{j,k}, C^*_k, D^*_k) \quad \text{where} \quad q_{k+1} \in C^*_k \tag{7}$$

As the agent is powered by strong MLLM $f_\theta$, we leverage its multimodal understanding capability to analyze each question, providing its reasoning process by comprehensively considering previous interview history $\mathcal{M}_k$, the real-time model capability $a_{j,k}$, the detailed semantics of candidate questions $C^*_k$ and the corresponding difficulties of candidate questions $D^*_k$. Ultimately, the agent selects the most appropriate question $q_{k+1}$ from these five candidates to serve as the next evaluation question, and we update the previously attempted question set as $Q'_{k+1} \leftarrow Q'_k \cup \{q_{k+1}\}$. Detailed prompts used for the interviewer agent are provided in Appendix I.

## 3.5 Dynamic Memory Update

To maintain contextual coherence during evaluation, AutoJudger supports the memory mechanism. Emphasizing long-term statistical awareness, the memory $\mathcal{M}$ accumulates high-level information about previously selected questions and model responses, grouped by semantically inferred categories as a markdown table. Since many benchmarks lack predefined class labels or contain noisy annotations, AutoJudger assigns categories based on semantic features and dynamically expands the category table as new topics emerge. For each category, the memory tracks statistics including the number of questions, max/min/average difficulty, and overall accuracy. This enables the agent to maintain global awareness of coverage and balance across domains. A representative example is:

| Category | Count | Max Difficulty | Min Difficulty | Avg Difficulty | Accuracy |
|----------|-------|----------------|----------------|----------------|----------|
| Accounting | 5 | 5.21 | -1.02 | 1.15 | 0.60 |
| Art History | 20 | 1.01 | -5.20 | -0.83 | 0.71 |
| Botany | 9 | 0.45 | -5.30 | -1.31 | 0.56 |
| Cell Biology | 14 | 4.90 | -2.44 | -0.70 | 0.50 |

This memory table illustrates a more realistic usage scenario, featuring a broad range of question difficulties and imbalanced category distributions. For example, Art History contains 20 questions spanning a wide difficulty range with relatively high accuracy, while Accounting tend to be more difficult with greater outcome variance. Such statistical tracking allows the agent to identify under-represented or overly challenging areas, informing more targeted selection in subsequent iterations.

## 4 Experiment

### 4.1 Experiment Setups

**Benchmarks** We validate the effectiveness of AutoJudger on four commonly-adopted benchmarks: MMMU-Dev&Val (Yue et al., 2024), SEEDBench-Image (Li et al., 2024b), MMT-Bench-Val (Ying et al., 2024), and AI2D-Test (Kembhavi et al., 2016). AI2D represents a relatively simple scenario, while the other three benchmarks are used for comprehensive multi-dimensional evaluation, simulating complex environment with a diverse question pool.

**Metrics** We propose a metric, ranking accuracy, to quantitatively assess the consistency between our evaluation results and the results on the full benchmark, defined as:

$$\rho = \text{Ranking Accuracy}(\%) = \left(1 - \frac{\#\text{Inversions}}{n * (n-1)/2}\right) * 100 \tag{8}$$

where the number of inversions refers to pairwise discrepancies between the predicted and ground-truth rankings, where the ground-truth ranking is determined by the model accuracy over the full benchmark. In addition to consistency, efficient benchmarking methods are supposed to be stable. We also report the confidence intervals (1.96 times the standard deviation) of the ranking accuracy based on multiple experiments. A narrower interval indicates the corresponding method is more stable.

Table 1: **Performance of different methods under 5% compression ratio**. We report the average ranking accuracy together with the confidence intervals. The best results are highlighted in **bold**.

| Method | AI2D$_{TEST}$ | MMMU$_{DEV\&VAL}$ | MMT-Bench | SEEDBench$_{IMG}$ |
|---|---|---|---|---|
| Random | 93.82±3.71 | 81.47±6.28 | 85.88±6.14 | **92.65±4.74** |
| Cluster | 93.53±3.80 | 78.97±11.62 | 87.79±3.71 | 92.50±3.58 |
| Stratified | 93.97±3.08 | 84.26±6.22 | 84.12±6.61 | 90.88±3.82 |
| IRT | 89.71±0.00 | 82.35±0.00 | 88.24±0.00 | 91.91±0.00 |
| **AutoJudger** | **94.85±0.00** | **87.94±0.71** | **92.06±1.41** | 90.74±0.71 |

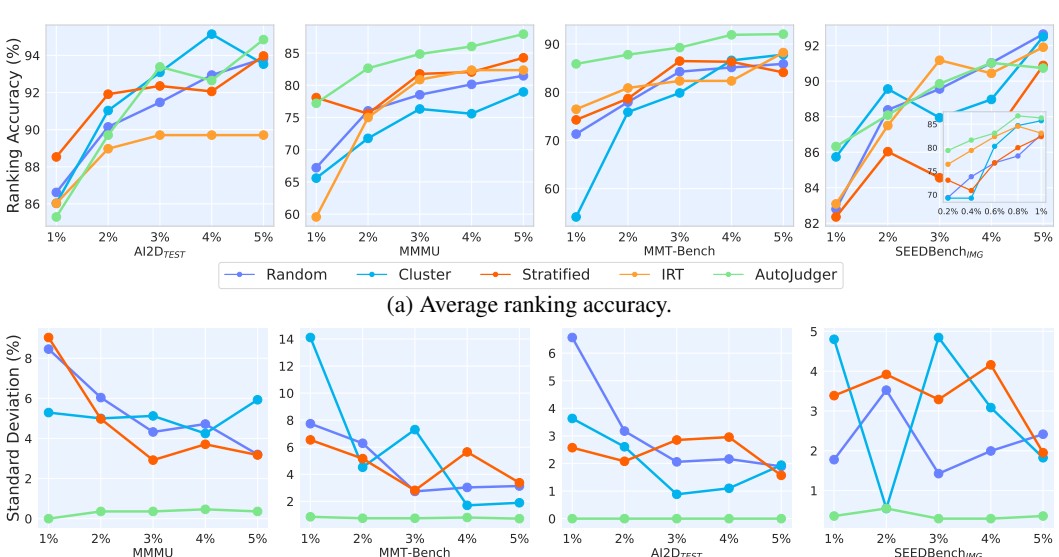

(a) Average ranking accuracy.

(b) Standard deviation of ranking accuracy.

Figure 3: **Evaluation performance and stability under varying compression ratios.**

**Baselines** We compare AutoJudger against a set of baseline strategies categorized into two groups: unified sampling and model-specific sampling. Unified sampling methods use a shared question pool for all models, including *Random Sampling (Random)*, which selects a fixed number of questions at random; Stratified Random Sampling (Stratified) (Perlitz et al., 2023) also selects a fixed number of questions randomly, but applies weighted sampling based on the number of categories within each benchmark; and *Cluster-Based Sampling (Cluster)*, which applies K-means clustering to the BERT embeddings of questions and selects those closest to each cluster centroid as the evaluation questions. In contrast, model-specific sampling strategies adapt question selection to individual models. The most representative method is *Optimal IRT Difficulty Choosing (IRT)* (Lord, 2012), which iteratively selects questions whose IRT-estimated difficulty is closest to the model's latent ability until the desired sample size is reached. Appendix D provides a detailed introduction to baseline methods.

**Implementation Details** We deploy our AutoJudger framework based on the Qwen2.5-VL-7B-Instruct model (Bai et al., 2025) where the retrieval module is driven by CLIP ViT-B/32 (Radford et al., 2021). For IRT-based difficulty assessment of the questions, we collect offline evaluation results from 60 models (training set)[1]. During evaluation, we used AutoJudger to assess another 17 models (test set). Both subsets of models cover a wide range of parameter scales, including both open-source and proprietary models, with no overlap between them. Please refer to Appendix D for more details. Each experiment is repeated five times to reduce the impact of randomness. We conduct all experiments on a Linux machine running Ubuntu 22, with 8 NVIDIA RTX 4090 GPUs.

### 4.2 MAIN RESULTS

In Table 1, we compare our framework with several efficient benchmarking baselines using only 5% of the data across four widely-used benchmarks. Three key observations emerge: (1) AutoJudger

---

[1]Offline results are collected from VLMEvalKit (Duan et al., 2024): `https://github.com/open-compass/VLMEvalKit`

Table 2: **Impact of different framework design of AutoJudger on four benchmarks.**

| Method Variant | AI2D$_{TEST}$ | MMMU$_{DEV\&VAL}$ | MMT-Bench | SEEDBench$_{IMG}$ |
|---|---|---|---|---|
| **AutoJudger** | **94.85** | 88.24 | **93.38** | **91.18** |
| $w/o$ agent | 92.28 | 82.87 | 86.62 | 91.10 |
| $w/o$ visual | 94.85 | 87.50 | 91.18 | 91.18 |
| $w/o$ memory | 94.12 | **89.71** | 89.71 | 91.18 |

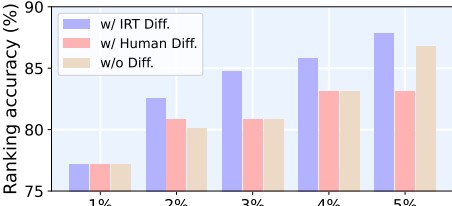
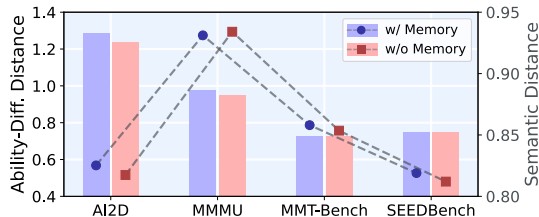

Figure 4: **Ranking accuracy of AutoJudger computed at MMMU$_{DEV\ VAL}$ under three distinct difficulty settings.**

Figure 5: **Comparison of ability-difficulty distance** (bar chart, left y-axis) **and semantic distance** (line plot, right y-axis) **with and without memroy.**

consistently outperforms all baselines on most benchmarks, demonstrating strong effectiveness in low-data regimes. (2) Compared to stochastic baselines (e.g., Random, Cluster, Stratified), AutoJudger exhibits significantly lower variance, indicating enhanced stability and robustness. (3) The IRT baseline, being a deterministic method, does not introduce randomness, but it suffers from suboptimal performance–for example, IRT performs notably worse on MMMU. In contrast, AutoJudger integrates real-time model feedback and history records to dynamically select appropriate questions. This adaptive strategy gradually mitigates the negative effects of randomness, improving the stability while achieving superior overall performance. Beyond the general datasets, we further explore applying AutoJudger to reasoning-oriented benchmarks, with results provided in Appendix F.

As illustrated in Figure 3, we compare methods across varying compression ratios. (1) **Accuracy**: While all baselines achieve good performance on AI2D, a relatively simple benchmark, AutoJudger exhibits consistent and substantial advantages on more complex benchmarks such as MMT-Bench and MMMU. Its smaller gains on SEEDBench can be attributed to the dataset's larger scale – about four times that of the others – where even 5% provides sufficient data for baseline convergence. At lower compression rates (<1%), however, AutoJudger's advantage becomes markedly pronounced, demonstrating reliable evaluation of complex benchmarks at significantly reduced cost. (2) **Stability**: Baseline methods require large subsets to ensure stability, whereas AutoJudger's adaptive strategy effectively maintains strong stability across different data scales. Besides different compression ratios, Appendix E investigates the effects of multiple factors (e.g. randomness in initialization) on AutoJudger, to demonstrate its stability and generalization capability.

### 4.3 EFFECTIVENESS OF THE AUTOJUDGER FRAMEWORK

Moreover, we conduct experiments to validate the effectiveness of core designs within AutoJudger.

**Necessity of Question Difficulty Estimation.** To investigate the necessity of providing question difficulty as reference information during the agent-driven next question selection, we conduct experiments on the MMMU dataset, which includes human-annotated difficulty labels. We compare three difficulty settings: (1) using IRT-estimated difficulty, (2) using human-annotated difficulty, and (3) excluding difficulty information entirely. Results in Figure 4 illustrate that applying IRT-estimated difficulty outperforms the other two settings across various compression ratios, indicating that the difficulty information is crucial for adaptive evaluation. However, manually-assessed difficulty may not align with the difficulty perceived by existing MLLMs.

**Necessity of Agent-based Question Selection.** To validate the necessity of agent-driven question selection, we conduct an ablation study by removing the agent from the AutoJudger framework. In this case, candidate questions are filtered using Equation 5, and a weighted sampling mechanism is performed based on the proximity between question difficulty and the model's estimated ability. As shown in the second line of Table 2, removing the agent leads to a significant performance drop in three datasets. The effect on SeedBench is less significant due to its large size, which causes different methods to converge. However, when the compression ratio is reduced to 1%, the ranking accuracy

Table 3: **Mean semantic distance between questions under 5% compression ratio.**

| Method | AI2D$_{TEST}$ | MMMU$_{DEV\&VAL}$ | MMT-Bench | SEEDBench$_{IMG}$ |
|---|---|---|---|---|
| Random | 0.8169 | 0.7534 | 0.7446 | 0.7359 |
| Cluster | 0.7661 | 0.7534 | 0.7549 | 0.7402 |
| IRT | 0.8198 | 0.7568 | 0.7417 | 0.7451 |
| AutoJudger | **0.9385** | **0.8149** | **0.8594** | **0.8262** |

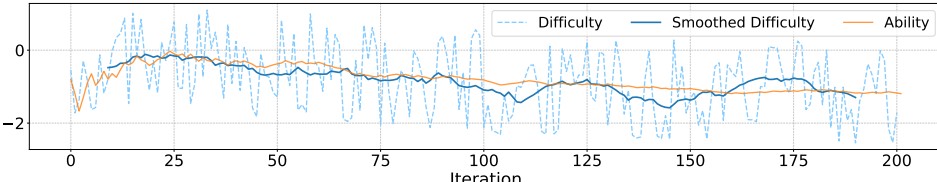

Figure 6: **The evolution of the estimated MiniCPM-V-2's ability and the question difficulty over the course of evaluation on MMMU$_{DEV\ VAL}$.** "Smoothed difficulty" is the average difficulty of the 20 nearest questions, while "difficulty" is the difficulty of the question selected at each iteration.

drops from 86.32% (with agent) to 84.56% (without agent). This demonstrates that the agent as the judger plays a crucial role in improving evaluation quality, especially in low-data scenarios.

**Necessity of using Visual Information.** To assess the importance of multimodal understanding in AutoJudger's decision-making process, we investigate the role of visual information in evaluation. We construct an ablated method where visual information is excluded from the context provided to the agent, offering only textual information instead. As presented in Table 2, removing visual information consistently harms the performance, indicating that the semantics of images are crucial for ensuring the diversity of selected questions in multimodal benchmarks.

**Necessity of Dynamic Memory $\mathcal{M}$.** To understand the contribution of the proposed dynamic memory $\mathcal{M}_k$, we analyze the impact of removing it from the framework. The ablated framework keeps the memory of AutoJudger empty, which transforms Equation equation 4 into $q_{k+1} = f\left(\varnothing, a_{j,k}, \mathcal{C}_k^*\right)$. As presented in the third line in Table 2, the performance degrades on AI2D and MMT-Bench. We hypothesize that without memory $\mathcal{M}$, AutoJudger relies solely on the estimated model ability $a_j$ for question selection, leading it to favor questions whose difficulty closely matches that ability. To verify our hypothesis, we compute the averaged absolute distance $\left|d_{q_{k+1}} - a_{j,k}\right|$ between the difficulty of selected question $q_{k+1}$ and the estimated ability $a_{j,k}$ of model $m_j$. As shown in the bar chart of Figure 5 (left y-axis), the ability-difficulty distance significantly decreases when memory is absent. We also compute the semantic distances between questions selected by AutoJudger. As illustrated by the line plot in Figure 5 (right y-axis), semantic diversity is higher when memory is present. These findings highlight the importance of memory in preserving a global view of model strengths and weaknesses, enabling more balanced and informative question selection.

### 4.4 COMPUTATIONAL OVERHEAD OF AUTOJUDGER

While AutoJudger substantially reduces the number of evaluation queries by selecting only the most informative ones, it inevitably introduces additional computational overhead. Taking that into consideration, we display the computational cost in Table 4 (see details in Appendix H).

- **Evaluating a 7B model**: One AutoJudger iteration incurs about 2.68× the cost of Qwen2.5-VL-7B model forward pass. However, since AutoJudger achieves high accuracy with only 4% of the full dataset, this translates to a relative cost of 10.7% compared to full-scale evaluation.
- **For larger or smaller models**: As evaluated model size changes, the evaluated model's inference cost adjusts, while the cost of AutoJudger remains fixed. The relative computational cost on evaluating 3B and 72B models is 19.4% and 4.7% respectively. The advantage is amplified when the evaluated model uses CoT reasoning or when external evaluators (e.g., GPT-4) are invoked for assessment — both of which add significant per-step overhead that AutoJudger avoids by design.

### 4.5 AUTOJUDGER STRIKES A BALANCE BETWEEN DIFFICULTY AND SEMANTIC DIVERSITY

AutoJudger is designed to select questions that both align with the model's ability and exhibit maximal diversity. To validate this, we first investigate the relationship between the estimated model's ability and the difficulty of recommended questions. As illustrated in Figure 6, AutoJudger adaptively selects

Table 4: **Computational overhead of AutoJudger.** $F_{\text{model}}$ is the FLOPs of a single forward pass of the evaluated model. $F_{\text{step}}$ is the per-step computational cost when evaluated with AutoJudger. $R(0.04)$ is the relative computational cost under 4% of the data.

| Model | $F_{\text{model}}$ | $F_{\text{step}}$ | $F_{\text{step}}/F_{\text{model}}$ | $R(0.04)$ |
|---|---|---|---|---|
| Qwen2.5-VL-3B | 6.04 | 29.29 | $4.85\times$ | 19.4% |
| Qwen2.5-VL-7B | 13.85 | 37.10 | $2.68\times$ | 10.7% |
| Qwen2.5-VL-72B | 139.97 | 163.22 | $1.17\times$ | 4.7% |

questions whose difficulty values dynamically match the ability of the evaluated model. Beyond difficulty alignment, we then assess whether AutoJudger preserves semantic diversity in the selected questions. We quantify the semantic similarity among selected questions via the average Euclidean distance between the embeddings generated by CLIP ViT-B/32. As shown in Table 3, the questions selected by AutoJudger have a significantly higher average semantic distance than those from other methods, indicating that AutoJudger achieves superior semantic diversity.

Besides quantitative analysis, we provide several cases in Appendix G for qualitative analysis. These examples demonstrate that AutoJudger is capable of comprehensively analyzing both difficulty and semantics of questions, and making recommendations based on historical information.

## 5 RELATED WORK

### 5.1 EFFICIENT BENCHMARKING

As MLLMs advance, numerous large-scale benchmarks (Xu et al., 2024b; Liu et al., 2024f; Lu et al., 2023; 2022) have been proposed to assess their capabilities across diverse tasks, such as diagram reasoning (Kembhavi et al., 2016), expert-level subject reasoning (Yue et al., 2024) and so on (Ying et al., 2024; Li et al., 2023; 2024b). While these resources provide broad coverage, their scale makes the evaluation increasingly expensive, especially when proprietary models (e.g., ChatGPT) are employed for scoring (Liu et al., 2023; 2024d; Lu et al., 2023).

To mitigate this, efficient benchmarking aims to select smaller yet informative subsets that preserve evaluation fidelity. Existing approaches fall into two categories: feature-based and difficulty-based sampling. Feature-based methods reduce redundancy by ensuring semantic representativeness, such as stratified sampling by sub-scenarios (Perlitz et al., 2023) or grouping by model confidence (Vivek et al., 2023). Difficulty-based methods adapt to model performance, by leveraging IRT (Cai et al., 2016) (Zhuang et al., 2023a), clustering by difficulty (Polo et al., 2024), or integrating IRT with Glicko-2 ratings (Ding et al., 2024). Beyond specific perspectives, ***AutoJudger*** unifies semantic diversity and difficulty adaptiveness, enabling both efficient and adaptive benchmarking of MLLMs.

### 5.2 DIFFICULTY ESTIMATION

Most existing benchmarks lack explicit difficulty annotations, limiting the granularity of capability evaluation. Classical psychometric frameworks such as IRT (Cai et al., 2016) model the probability of a correct response as a function of latent question difficulty and subject ability. Traditionally used in standardized exams like the GRE and SAT (An & Yung, 2014), IRT has been extended to NLP settings to analyze benchmark saturation (Vania et al., 2021) and estimate model proficiency (Park et al., 2024). These applications demonstrate IRT's flexibility in both question-level diagnostics and model ranking, making it a natural fit for difficulty estimation in LLM evaluation. Other lines of works access difficulty via content-based features (Jiao et al., 2023; Xu et al., 2024a), step-level reasoning complexity (Cheng et al., 2021; Wang et al., 2024b), or LLM-based prediction models (Gao et al., 2018; Lee et al., 2023). In our work, we incorporate IRT into the adaptive benchmark construction process, where question difficulty is pre-estimated from historical model responses and then used to enable dynamic, targeted evaluations across varying model capabilities.

## 6 CONCLUSION

We propose AutoJudger to tackle the rising cost of evaluating MLLMs. Leveraging an agent-driven question selection framework, we demonstrate that it is possible to consistently assess the capabilities of MLLMs with only 5% samples in multimodal benchmarks. Extensive experiments illustrate that AutoJudger is not only effective but also stable under various evaluation settings. We believe AutoJudger is a promising solution for scalable and reliable MLLM evaluation.

## 7 ETHICS STATEMENT

All benchmark datasets and evaluation records used in this research are publicly available and properly cited. The experiments were conducted in accordance with the original dataset licenses and usage guidelines.

## 8 REPRODUCIBILITY STATEMENT

We provide all code, data, and instructions necessary to reproduce the results reported in this paper. The materials are available through the following anonymized link: `https://anonymous.4open.science/r/AutoJudger-anonymous`.

The repository includes:

- Source code for the AutoJudger framework.
- Data used in our experiments.
- Detailed instructions for running experiments and reproducing all reported results.

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

## A  THE USE OF LARGE LANGUAGE MODELS (LLMs)

LLMs were employed solely to polish the text and correct minor grammar issues.

## B  DETAILS OF RASCH MODEL (IRT) FITTING

To estimate the question difficulty vector $D$ from the binary response matrix $\{r_{ij}\}$, where $r_{ij} = 1$ indicates that model $m'_j$ correctly answers question $q_i$, we adopt the Rasch model—a one-parameter logistic IRT model as shown in Equation 3.

**Variational Bayesian Framework**  We use variational inference to approximate the posterior distribution over model abilities and question difficulties. Specifically, we assume a fully factorized variational distribution (Ding et al., 2024):

$$q(a, D) = \prod_j q(a_j) \prod_i q(d_i) \tag{9}$$

Each latent variable is modeled as a Gaussian:

$$q(a_j) = \mathcal{N}(\mu_{a_j}, \sigma_{a_j}^2), \quad q(d_i) = \mathcal{N}(\mu_{d_i}, \sigma_{d_i}^2) \tag{10}$$

The optimization target is the evidence lower bound (ELBO):

$$\mathcal{L}_{\text{ELBO}} = \mathbb{E}_{q(a,D)}[\log p(r \mid a, D)] - \text{KL}(q(a, D) \,\|\, p(a, D)) \tag{11}$$

We adopt standard Gaussian priors: $p(a_j) = \mathcal{N}(0, 1)$ and $p(d_i) = \mathcal{N}(0, 10^3)$, which yield closed-form KL divergences.

**Optimization**  We optimize the ELBO using stochastic gradient descent. Gradients are estimated via the reparameterization trick:

$$\begin{aligned} a_j &= \mu_{a_j} + \sigma_{a_j} \cdot \epsilon_j, \quad \epsilon_j \sim \mathcal{N}(0, 1) \\ d_i &= \mu_{d_i} + \sigma_{d_i} \cdot \epsilon_i, \quad \epsilon_i \sim \mathcal{N}(0, 1) \end{aligned} \tag{12}$$

This leads to efficient and low-variance updates for the variational parameters $\mu$ and $\sigma$.

**Implementation Setting**  We implement the model using PyTorch (Paszke et al., 2017) and Pyro (Bingham et al., 2017). The variational distributions over model abilities and question difficulties are initialized with zero mean and large variance. Specifically, the ability parameters $a_j$ are initialized with $\mu_{a_j} = 0$, $\sigma_{a_j} = 1$, while the difficulty parameters $d_i$ are initialized with $\mu_{d_i} = 0$, $\sigma_{d_i} = 10^3$, corresponding to vague priors that reflect minimal prior knowledge.

We optimize the ELBO (Kingma & Welling, 2022) using the Adam optimizer (Kingma & Ba, 2017) with a learning rate of 0.1 for 3,200 steps, using mini-batches sampled from the response matrix $\{r_{ij}\}$. Training terminates when the relative change in ELBO falls below $1 \times 10^{-4}$ within a moving window. During inference, we use the variational mean $\mu_{d_i}$ as the point estimate of question difficulty.

## C  DETAILS OF MODEL ABILITY ESTIMATION

To estimate the model ability $a_j$ based on its responses to a subset of questions, we employ a binary search algorithm grounded in the one-parameter logistic IRT (Rasch) model. Specifically, we solve the following maximum likelihood estimation problem:

$$\max_{a_j} \sum_{i \in Q'} \log p(r_{ij} \mid a_j, d_i), \tag{13}$$

where $p(r_{ij} \mid a_j, d_i)$ is defined in Equation 3. Since the log likelihood is a monotonic function with respect to $a_j$, we perform binary search within a bounded interval $[-30, 30]$, iteratively updating the estimate until convergence. The stopping criterion is based on a fixed threshold of $10^{-5}$ for either the log-likelihood difference or the change in $a_j$.

This procedure enables efficient and stable estimation of real-time model ability during evaluation, while keeping the question difficulties $\{d_i\}$ fixed.

## D   IMPLEMENTATION DETAILS OF EFFICIENT BENCHMARKING METHODS

### D.1   BASELINES

We detail the implementation of the baselines below.

- *Random Sampling (Random)*: We uniformly sample $\delta * |Q|$ questions from the complete evaluation benchmark $Q$ without replacement.

- *Stratified Random Sampling (Stratified)* (Perlitz et al., 2023): We partition the question pool based on provided category labels and draw approximately equal numbers of questions from each partition. We ensure that the maximum size difference between any two categories is no greater than one. Sampling is performed independently per category without replacement.

- *Cluster-Based Sampling (Cluster)*: Each question is embedded using the CLIP ViT-B/32 encoder (Radford et al., 2021), producing a 512-dimensional representation. Embeddings are L2-normalized before clustering. We apply K-means clustering to partition the question pool. The number of clusters $K$ is set to the number of desired questions, i.e., $K = \delta \cdot |Q|$. One question is selected per cluster, chosen as the one closest to the centroid in Euclidean space.

- *Optimal IRT Difficulty Choosing (IRT)* (Lord, 2012): We use a one-parameter logistic Item Response Theory model to adaptively select questions based on the model's estimated ability. The ability score is initialized with a simple prior: we assume the model has answered five medium-difficulty questions (difficulty 0) and got 2.5 correct on average. This initialization prevents unstable updates in early iterations.

### D.2   OUR FRAMEWORK: AUTOJUDGER

The evaluation workflow of AutoJudger is presented in Algorithm 1. All questions in each benchmark are first annotated with estimated difficulty levels. Then, evaluation begins with a standardized initialization, followed by iterative refinement of the question set based on the model's responses.

**Training and Test Models**   To ensure a representative and balanced evaluation, we partition the models based on parameter scale, as the model capability is generally observed to improve with increasing parameter size. Accordingly, we divide the models into four groups: $< 5B$, $< 9B$, $< 16B$, and $\geq 16B$ (including proprietary models). From each group, we randomly sample 20% of the models as test models, the remaining 80% are used as training models to collect offline responses which are utilized for question difficulty estimation. This stratified selection strategy ensures that AutoJudger is evaluated across a wide spectrum in terms of model abilities. The complete list of the 60 training models used for IRT-based question difficulty assessment are provided in Table 5 and the left 17 models evaluated via AutoJudger are listed in Table 6.

---

**Algorithm 1 AutoJudger** - Adaptive Ability Estimation via Multimodal Large Language Model

---

**Require:**
1: Question Pool $Q = \{q_i\}$, Testing Models $M = \{m_j\}$
2: An $Agent$ as the judger to select the question
3: **for all** test model $m_j \in M$ **do**
4:     Initialize problem assistant $Agent$
5:     Sample initial batch of questions $Q_0$
6:     collect model responses $R_0$
7:     Estimate initial model ability $a_0$.
8:     Generate init memory $\mathcal{M}_0$.
9:     **for** $t = 1$ to $(\delta \times |Q| - |Q_0|)$ **do**
10:         Get the candidate set $\mathcal{C}^*$
11:         $Agent$ select the next question with $\mathcal{M}_{t-1}, a_{t-1}$ given. $(q_t = f(\mathcal{M}_{t-1}, a_{t-1}, \mathcal{C}^*))$
12:         Get the answer $r_{tj}$, update the ability $a_t$
13:         Update $\mathcal{M}_t$ with $q_t$ included.
14:     **end for**
15:     Get the final ability estimation $a_{(\delta \times |Q| - |Q_0|)}$
16: **end for**

---

**Initialization Details**  In the initialization phase, we take the text of each question as input and use CLIP ViT-B/32 as the encoding model (this design choice is provided in Appendix E.1) to generate normalized vector representations. We aim to select a diverse and representative set of questions, so we apply k-means clustering with k=10 and select the questions closest (in terms of L2 distance) to each of the resulting cluster centers. To mitigate the instability of k-means, we repeat the clustering process 50 times and choose the set that achieves the highest ranking accuracy on the training set.

# E  FURTHER ANALYSIS ON GENERALIZATION AND STABILITY OF AUTOJUDGER ACROSS DIFFERENT FACTORS

In this section, we investigate how different factors affect the performance of AutoJudger. Overall, this section demonstrates AutoJudger's stability and generalization across various factors, including: robustness to random initialization (E.1); generalization under different question selection strategies and even in the absence of historical problem-solving records (E.2); generalization across different agent backbones (E.3); and generalization across different IRT settings (E.4).

## E.1  THE IMPACT OF INITIALIZATION

Initialization is a crucial component of the AutoJudger framework, as it determines the starting point for iterative evaluation. To systematically investigate its impact, we conduct experiments to compare the performance of different initialization methods, including two baselines: random sampling and sampling based on question difficulty quantiles (denoted as IRT), as well as our clustering-based initialization. Additionally, we compare the performance of different embedding strategies, including the use of different encoders and different semantic representations of questions.

To reduce the impact of randomness, each configuration is evaluated over 50 independent trials, and the average ranking accuracy is reported. To avoid information leakage and overfitting to the test set, we report the ranking accuracy on the training-set models. As shown in Table 7, there are several findings: (1) Incorporating additional information—whether related to difficulty or semantics—through appropriate methods improves the effectiveness of initialization. (2) While approaches based on difficulty perform well, using semantic information yields the best results. (3) Unlike Qwen2.5-VL, the dual-encoder CLIP model struggles to integrate information from multiple modalities. (4) Textual features reflect the diversity of questions more effectively than visual features, suggesting that current benchmarks may not consider the richness of visual information.

## E.2  THE IMPACT OF CANDIDATE QUESTION RETRIEVAL STRATEGY

**Superiority of Personalized Retrieval**  As argued in the introduction, we believe each model should be assigned with a personalized evaluation subset since models vary in capability. For instance, evaluating powerful models with too many easy questions may provide limited information. To validate this argument, we conduct an experiment to assess top-performing models (top 50% in terms

Table 5: **List of 60 training set models used for IRT-based question difficulty assessment.** These models span a range of sizes and include both open-source and proprietary models.

| Models | Open-source | # Params (B) | Date |
|---|---|---|---|
| InternVL2-1B (Chen et al., 2024c) | Yes | 0.9 | 2024.11 |
| llava-onevision-qwen2-0.5B-ov (Li et al., 2024a) | Yes | 0.9 | 2024.07 |
| llava-onevision-qwen2-0.5B-si (Li et al., 2024a) | Yes | 0.9 | 2024.07 |
| h2ovl-mississippi-1B (Galib et al., 2024) | Yes | 0.8 | 2024.01 |
| NVLM (Dai et al., 2024) | Yes | 79.4 | 2024.09 |
| Qwen2-VL-72B-Instruct (Wang et al., 2024a) | Yes | 72 | 2024.08 |
| 360VL-70B (Qihoo360 AI Lab, 2024) | Yes | 71 | 2024.04 |
| InternVL2-40B (Chen et al., 2024c) | Yes | 40.1 | 2024.06 |
| InternVL-Chat-V1-5 (Chen et al., 2024c) | Yes | 25.5 | 2024.03 |
| InternVL2-26B (Chen et al., 2024c) | Yes | 25.5 | 2024.11 |
| MMAlaya2 (Ltd., 2024) | Yes | 25.5 | 2024.08 |
| Eagle-X5-13B (Shi et al., 2024) | Yes | 15.4 | 2024.08 |
| Slime-13B (Zhang et al., 2024a) | Yes | 13.4 | 2024.05 |
| TransCore-M (PCIResearch, 2023) | Yes | 13.4 | 2024.03 |
| llava-v1.5-13B (Liu et al., 2024b) | Yes | 13 | 2024.01 |
| Falcon2-VLM-11B (Technology Innovation Institute (TII), 2024) | Yes | 11 | 2024.07 |
| Ovis1.6-Gemma2-9B (Lu et al., 2024c) | Yes | 10.2 | 2024.09 |
| monkey (Li et al., 2024e) | Yes | 9.8 | 2023.11 |
| monkey-chat (Li et al., 2024e) | Yes | 9.8 | 2023.11 |
| POINTS-Yi-1.5-9B-Chat (Liu et al., 2024e) | Yes | 9.5 | 2024.09 |
| Mantis-8B-Fuyu (Jiang et al., 2024b) | Yes | 9.4 | 2024.04 |
| Eagle-X5-7B (Shi et al., 2024) | Yes | 9.1 | 2024.08 |
| Bunny-llama3-8B (He et al., 2024) | Yes | 8.5 | 2024.04 |
| Mantis-8B-siglip-llama3 (Jiang et al., 2024a) | Yes | 8.5 | 2024.04 |
| Mantis-8B-Idefics2 (Jiang et al., 2024a) | Yes | 8.4 | 2024.05 |
| Slime-8B (Zhang et al., 2024b) | Yes | 8.4 | 2024.05 |
| llava-next-llama3 (Liu et al., 2024c) | Yes | 8.3 | 2024.04 |
| POINTS-Qwen-2.5-7B-Chat (Liu et al., 2024e) | Yes | 8.3 | 2024.12 |
| llava-next-interleave-7B (Liu et al., 2024c) | Yes | 8.1 | 2024.06 |
| llava-next-interleave-7B-dpo (Liu et al., 2024c) | Yes | 8.1 | 2024.06 |
| MiniCPM-V-2-6 (Yao et al., 2024) | Yes | 8.1 | 2024.07 |
| InternVL2-8B (Chen et al., 2024b) | Yes | 8.1 | 2024.11 |
| llava-onevision-qwen2-7B-ov (Li et al., 2024a) | Yes | 8.0 | 2024.07 |
| Ovis1.5-Llama3-8B (Lu et al., 2024c) | Yes | 8 | 2024.07 |
| molmo-7B-O-0924 (Deitke et al., 2024a) | Yes | 7.7 | 2024.09 |
| llava-next-mistral-7B (Liu et al., 2024c) | Yes | 7.6 | 2024.03 |
| deepseek-vl-7B (Lu et al., 2024a) | Yes | 7.3 | 2024.02 |
| llava-next-vicuna-7B (Liu et al., 2024c) | Yes | 7.1 | 2024.05 |
| XComposer2 (Dong et al., 2024) | Yes | 7 | 2024.01 |
| llava-v1.5-7B (Liu et al., 2024b) | Yes | 7 | 2024.01 |
| Phi-3-Vision (Abdin et al., 2024) | Yes | 4.2 | 2024.05 |
| InternVL2-4B (Chen et al., 2024b) | Yes | 3.7 | 2024.11 |
| Vintern-3B-beta (Doan et al., 2024) | Yes | 3.2 | 2024.01 |
| BlueLM-V (Lu et al., 2024d) | No | 3 | 2024.11 |
| paligemma-3B-mix-448 (Beyer* et al., 2024) | Yes | 2.9 | 2024.04 |
| InternVL2-2B (Chen et al., 2024b) | Yes | 2.2 | 2024.11 |
| Aquila-VL-2B (Gu et al., 2024) | Yes | 2.2 | 2024.01 |
| deepseek-vl-1.3B (Lu et al., 2024b) | Yes | 2.0 | 2024.02 |
| Moondream1 (vikhyatk, 2024) | Yes | 1.9 | 2024.01 |
| XComposer2-1.8B (Dong et al., 2024) | Yes | 1.8 | 2024.01 |
| Kosmos2 (Peng et al., 2023) | Yes | 1.7 | 2023.06 |
| molmoE-1B-0924 (Deitke et al., 2024b) | Yes | 1 | 2024.09 |
| GPT4V-20240409-HIGH (OpenAI, 2023) | No | - | 2024.04 |
| GPT4o (Hurst et al., 2024) | No | - | 2024.05 |
| GPT4o-HIGH (Hurst et al., 2024) | No | - | 2024.05 |
| GeminiFlash1-5 (Team et al., 2024) | No | - | 2024.09 |
| JT-VL-Chat (Corporation, 2024) | No | - | 2024.10 |
| Qwen-VL-Max-0809 (Bai et al., 2023) | No | - | 2024.08 |
| Qwen-VL-Plus-0809 (Bai et al., 2023) | No | - | 2024.08 |
| Taiyi (Luo et al., 2024) | No | - | 2023.11 |

of average ranks), either with their personalized questions picked by AutoJudger or simple questions that are selected to evaluate the worst model.

Results are provided in Table 8. Considering the efficiency, SEEDBench is excluded in this experiment due to its large scale. Since AI2D is a relatively easy scenario (minimal variation in difficulty across the questions), the "simplest" strategy demonstrates comparative performance. However, on more complex benchmarks like MMT and MMMU, the personalized approach demonstrates superior performance. Additionally, we observed that the "simplest" strategy lacks stability and does not necessarily improve as the dataset size increases. Generally, by dynamically selecting questions tailored to each model's capability, the proposed AutoJudger framework better accommodates varying model strengths and avoids overfitting to the preference of specific models. Therefore, we adopt the personalized strategy to retrieve questions.

**Candidate Question Selection Strategy** As stated in Equation 6 in Section 3.3, we select questions with largest semantic distance as the candidates ("semantic farthest"). To demonstrate the superiority of our approach, we compare it against two widely adopted question selection baselines: random sampling ("random"), and selecting questions with the smallest difficulty distance ("optimal difficulty"). As summarized in Table 9, while the "optimal difficulty" strategy achieves the best performance on

Table 6: **List of 17 test set models evaluated using AutoJudger.** These models are disjoint from the training set and representative in terms of diverse types and scales.

| Models | Open-source | # Params (B) | Date |
|---|---|---|---|
| InternVL2-76B (Chen et al., 2024a) | Yes | 76.3 | 2024.06 |
| llava-next-vicuna-13B (Liu et al., 2024c) | Yes | 13.4 | 2024.02 |
| Pixtral-12B (Agrawal et al., 2024) | Yes | 12 | 2024.08 |
| Ovis1.5-Gemma2-9B (Lu et al., 2024c) | Yes | 11.4 | 2024.07 |
| idefics2-8B (Laurençon et al., 2024) | Yes | 8.4 | 2024.03 |
| Mantis-8B-clip-llama3 (Jiang et al., 2024b) | Yes | 8.3 | 2024.01 |
| llava-onevision-qwen2-7B_si (Li et al., 2024a) | Yes | 8.0 | 2024.07 |
| molmo-7B-D-0924 (Deitke et al., 2024a) | Yes | 8.0 | 2024.09 |
| Slime-7B (Zhang et al., 2024a) | Yes | 7.1 | 2024.05 |
| Ovis1.6-Llama3.2-3B (Lu et al., 2024c) | Yes | 4.1 | 2024.01 |
| MiniCPM-V-2 (Yao et al., 2024) | Yes | 3.4 | 2024.11 |
| h2ovl-mississippi-2B (Galib et al., 2024) | Yes | 2.1 | 2024.01 |
| Janus-1.3B (Wu et al., 2024) | Yes | 2.1 | 2024.01 |
| Moondream2 (Korrapati, 2024) | Yes | 1.9 | 2024.02 |
| GPT4o-20240806 (Hurst et al., 2024) | No | - | 2024.08 |
| GeminiPro1-5 (Team et al., 2024) | No | - | 2024.09 |
| Step1V (StepFun, 2024) | No | - | 2024.03 |

Table 7: **Comparison between different initialization methods.** "Multi-concat" and "multi-mean" refer to concatenating and averaging the image and text embeddings, respectively, to serve as the multi-modal representations of questions. We report the average rank of each method (among all methods) across four benchmarks as the overall performance.

| Encoding Models | Input | AI2D$_{TEST}$ | MMMU$_{DEV\&VAL}$ | MMT-Bench | SEEDBench$_{IMG}$ | Avg Rank |
|---|---|---|---|---|---|---|
| CLIP ViT-B/32 | multi-concat | 74.98 | 72.54 | 61.81 | 58.00 | 7.75 |
|  | multi-mean | 72.61 | 70.57 | 55.07 | 55.92 | 10.00 |
|  | image | 69.80 | 64.72 | 66.07 | 65.45 | 7.50 |
|  | text | 81.31 | 73.02 | 60.88 | 75.59 | 3.75 |
| CLIP ViT-L/14 | multi-concat | 67.23 | 75.53 | 52.00 | 61.61 | 9.50 |
|  | multi-mean | 75.70 | 75.79 | 67.00 | 54.88 | 5.25 |
|  | image | 73.24 | 74.52 | 62.30 | 64.01 | 6.00 |
|  | text | 77.45 | 66.38 | 66.15 | 62.62 | 6.25 |
| Qwen2.5-VL-7B | text&image | 78.12 | 62.77 | 62.02 | 67.91 | 6.25 |
|  | image | 70.77 | 64.77 | 63.55 | 64.53 | 8.00 |
|  | text | 79.58 | 66.15 | 59.82 | 64.53 | 7.25 |
| IRT | - | 80.40 | 70.40 | 61.81 | 64.75 | 5.00 |
| Random | - | 74.62 | 68.83 | 60.36 | 63.33 | 8.50 |

the AI2D$_{TEST}$ benchmark, its effectiveness does not generalize well across other benchmarks. In contrast, the "semantic farthest" strategy demonstrates consistently strong performance across all evaluated benchmarks and under different compression ratios. Therefore, we choose to use semantic farthest strategy, as it not only exhibits broad applicability and consistent performance across diverse benchmarks, but can also introduce greater informational diversity.

**Expansion of Candidate Question Pool** We investigate the impact of the number of candidate questions (see Equation 6) on the performance of AutoJudger by expanding $|C_k^*|$ from 5 to 7 and 10. As shown in Table 10, a larger candidate set introduces more flexibility, but also brings additional noise, making it harder to identify the optimal next question.

**Evaluation without Historical Records** AutoJudger leverages item-level difficulty estimates and historical model response records to guide question selection. To examine its robustness when historical records are unavailable (e.g., in benchmarks expanded with new tasks), we conducted an ablation study comparing standard AutoJudger with a variant that does not use difficulty information. In the difficulty-agnostic variant ($w/o$ difficulty), all questions are assumed to have the same difficulty level. Table 11 summarizes the results across multiple benchmarks. The results show that even without explicit difficulty information, AutoJudger maintains competitive performance, though with

Table 8: **Ranking accuracy for top-performing models with personalized and unified question selection strategies.** "personalized" means the questions are selected via Equation 5. "simplest" means the questions are from the evaluation of the lowest-ranked model and fixed for all models.

| Benchmark | Questions | Compression Ratio | | | | |
|---|---|---|---|---|---|---|
| | | 1% | 2% | 3% | 4% | 5% |
| AI2D$_{TEST}$ | personalized | **95.24** | **95.24** | 95.24 | **95.24** | **95.24** |
| | simplest | 85.71 | 90.48 | **100.0** | 95.24 | 90.48 |
| MMMU$_{DEV\&VAL}$ | personalized | **78.57** | **81.25** | **82.14** | **88.39** | **88.39** |
| | simplest | **78.57** | 80.00 | 80.71 | 85.71 | 85.71 |
| MMT-Bench | personalized | **86.61** | **86.61** | **90.18** | **92.86** | **93.75** |
| | simplest | 85.71 | 82.14 | 78.57 | 65.71 | 67.14 |

Table 9: **Comparison of different candidate question selection strategies.** "Semantic farthest" means selecting questions with the largest semantic distance, "optimal difficulty" means selecting questions with the smallest difficulty distance, and "random" means purely random selection.

| Benchmark | Strategy | Compression Ratio | | | | |
|---|---|---|---|---|---|---|
| | | 1% | 2% | 3% | 4% | 5% |
| AI2D$_{TEST}$ | semantic farthest | 85.29 | 89.71 | 93.38 | 92.65 | 94.85 |
| | optimal difficulty | **90.44** | **93.38** | **93.38** | **94.12** | **95.59** |
| | random | 86.76 | 91.18 | 91.18 | 91.91 | 91.18 |
| MMMU$_{DEV\&VAL}$ | semantic farthest | 77.21 | 82.54 | 84.74 | **85.85** | **87.94** |
| | optimal difficulty | 77.21 | 77.21 | 78.68 | 80.15 | 83.09 |
| | random | 77.21 | **83.09** | **86.03** | 83.82 | 83.82 |
| MMT-Bench | semantic farthest | **85.66** | **87.87** | **89.34** | **91.91** | **92.06** |
| | optimal difficulty | 72.06 | 78.68 | 78.68 | 83.09 | 82.35 |
| | random | 77.94 | 83.82 | 88.97 | 86.76 | 91.18 |
| SEEDBench$_{IMG}$ | semantic farthest | **86.40** | **88.24** | **89.89** | **90.99** | **90.74** |
| | optimal difficulty | 72.06 | 78.68 | 78.68 | 83.09 | 82.35 |
| | random | 78.68 | 83.82 | 81.68 | 83.82 | 88.97 |

some degradation on more challenging benchmarks. This suggests that the framework is robust to the absence of historical response data.

### E.3 THE IMPACT OF FOUNDATION MODELS ON AUTOJUDGER

**Scaling up the Judging Agent** As the capability of the interviewer/judging agent plays a crucial role in our framework. We first explore to replace the original 7B-scale agent with a larger backbone, i.e. Qwen2.5-VL-32B-Instruct, to assess the impact of the scale of judging agent. Figure 7 presents a comparison between the 7B and 32B models on SEEDBench under varying compression ratios. The results show that the 32B model consistently outperforms the 7B models, especially in low-ratio settings. For example, at a compression ratio of 0.2%, the 32B model achieves a ranking accuracy of 82.7%, representing a 3.3% improvement over the 7B model. Although both models improve as the compression ratio increases, the 32B model remains consistently stronger, demonstrating better generalization and selection capability.

**Generalization to Different Backbones** We further evaluate AutoJudger with diverse judging backbones to determine whether its performance gains arise from the framework itself rather than a specific model. Table 12 reports results across representative benchmarks. In all cases, AutoJudger consistently surpasses the strongest baseline. For example, with Qwen2.5-VL-7B, InternVL2.5-8B, and GPT-4o-mini, AutoJudger improves accuracy on MMMU by up to 3.7% compared to the best baseline. The above findings indicate that the AutoJudger framework not only demonstrates strong performance, but also shows potential for further expansion. A more advanced interviewer agent could further enhance the effectiveness of AutoJudger.

Table 10: **The impact of different number of candidate questions.**

| Benchmark | # Candidate | Compression Ratio | | | | |
|---|---|---|---|---|---|---|
| | | 1% | 2% | 3% | 4% | 5% |
| $AI2D_{TEST}$ | 5 | 85.29 | 89.71 | **93.38** | 92.65 | **94.85** |
| | 7 | **88.24** | **92.65** | **93.38** | 93.38 | **94.85** |
| | 10 | 83.82 | 86.03 | 89.71 | 91.18 | 91.91 |
| $MMMU_{DEV\&VAL}$ | 5 | **77.21** | **82.54** | **84.74** | **85.85** | **87.94** |
| | 7 | **77.21** | 82.35 | 80.88 | 83.09 | 83.82 |
| | 10 | **77.21** | 82.35 | 83.82 | 85.29 | 85.29 |
| MMT-Bench | 5 | **85.66** | 87.87 | **89.34** | 91.91 | 92.06 |
| | 7 | 83.09 | **88.24** | 88.24 | 88.97 | **92.65** |
| | 10 | 82.35 | 86.03 | 88.24 | 91.18 | 91.18 |
| $SEEDBench_{IMG}$ | 5 | **86.40** | **88.24** | **89.89** | 90.99 | 90.74 |
| | 7 | 84.56 | 86.76 | 88.97 | **92.65** | **92.65** |
| | 10 | 86.03 | 86.76 | 88.24 | 88.24 | 88.97 |

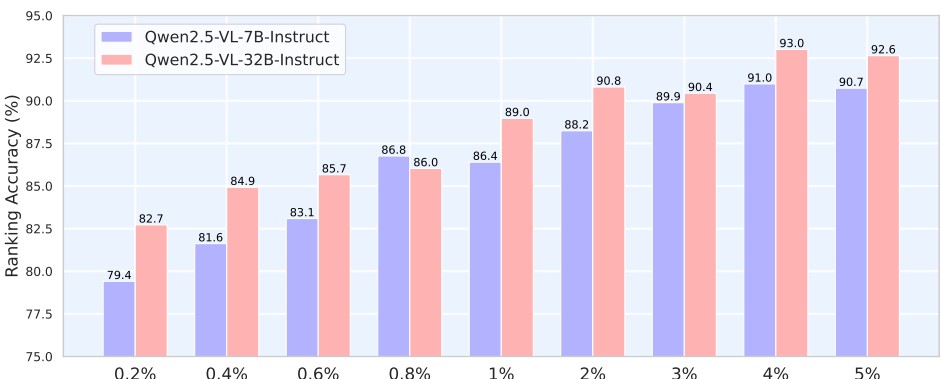

Figure 7: Comparison of 7B and 32B models on $SEEDBench_{IMG}$ at different compression ratios.

### E.4 THE IMPACT OF IRT MODELS

In AutoJudger, question difficulty is estimated using a 1PL model, combined with semantic features for item selection. To investigate whether incorporating additional IRT parameters improves performance, we consider more complex models: 2PL, which models item discrimination, and 3PL, which additionally accounts for guessing.

We conducted experiments on two representative benchmarks, and the results are summarized in Table 13. The results indicate that increasing the complexity of the IRT model does not consistently improve recommendation performance (AutoJudger still outperforms the strongest baseline across all settings). In these multi-modal, high-dimensionall settings, additional parameters may lead to overfitting or provide limited benefit. Therefore, the 1PL model offers a favorable balance between simplicity and effectiveness for AutoJudger.

## F EMPIRICAL EVIDENCE ON REASONING-ORIENTED BENCHMARKS

To further assess the robustness of AUTOJUDGER under reasoning-oriented benchmarks, we conducted evaluations on **MathVista-testmini**, a subset of MathVista designed for complex reasoning problems. As shown in Table 14, AutoJudger achieves the best performance, demonstrating strong generalizability in reasoning-intensive settings.

## G CASE STUDY

As an agent-driven evaluation framework, AutoJudger offers a major advantage in enhancing the interpretability of assessment results. We provide two representative examples in Figure 8 and

Table 11: Performance of AutoJudger with and without difficulty modeling.

| Method | AI2D$_{TEST}$ | MMMU$_{DEV\&VAL}$ | MMT-Bench | SEEDBench$_{IMG}$ |
|---|---|---|---|---|
| Baseline (best) | 93.97 | 84.26 | 88.24 | 92.65 |
| AutoJudger | **94.85** | **87.94** | **92.06** | **90.74** |
| $w/o$ difficulty | 94.85 | 86.76 | 89.71 | 88.97 |

Table 12: Performance of AutoJudger with different backbones.

| Method | AI2D$_{TEST}$ | MMMU$_{DEV\&VAL}$ | MMT-Bench | SEEDBench$_{IMG}$ |
|---|---|---|---|---|
| Baseline (best) | 93.97 | 84.26 | 88.24 | 85.74 |
| Qwen2.5-VL-7B | **94.85** | **87.94** | **92.06** | **86.32** |
| InternVL2.5-8B | 94.12 | 88.24 | 91.91 | 85.29 |
| GPT-4o-mini | – | – | 90.44 | – |

Figure 9. These cases illustrate how information stored in the dynamic memory enables the agent to efficiently analyze the evaluated model's performance across different types of questions (highlighted in blue text in the figures), thereby guiding more informed selection of subsequent evaluation items. Furthermore, the combination of model ability estimation and corresponding question difficulty analysis (marked in yellow and orange) assists the agent in identifying the most appropriate questions. Supported by these key components, AutoJudger can not only evaluate models efficiently, but also provide transparent reasoning behind each evaluation decision. We believe this is an essential step toward building trustworthy and transparent evaluation frameworks for future AI systems.

## H  DETAILS OF COMPUTATIONAL COST

According to the framework we have introduced in Section 3, the computational cost of AutoJudger can be divided into three parts: question difficulty estimation, initialization and iteration.

**Question Difficulty Estimation (pre-computed, negligible cost)**  Before the evaluation begins, AutoJudger estimates the difficulty of each benchmark question using offline evaluation results from a set of models. These response records are processed with Item Response Theory (IRT) to derive fixed difficulty scores. Since this procedure is performed entirely offline and does not recur during the actual evaluation, its cost is negligible and excluded from the runtime computation overhead.

**Initialization (one-time cost)**  At the start of evaluation, as the ability of the model is unknown, AutoJudger constructs an initial question pool to build a strong starting point. This involves:

- Autojudger computing semantic embeddings for all questions (e.g., via CLIP),
- Autojudger performing similarity computation and clustering,
- Autojudger sampling a diverse subset of $\beta$ questions to bootstrap ability estimation,
- the evaluated model solving the selected questions (via forwad pass), and
- Autojudger generating an intial summary and memory table.

The cost of this one-time procedure is fixed and denoted as $F_{\text{init}}$. Let us take AutoJudger built upon Qwen2.5-VL-7B to conduct evaluations on the MMT benchmark as an example. The initialization cost involves encoding the CLIP embeddings of questions (6.06 TFLOPs), calculating the pairwise similarity between these questions (15.02 GFLOPs), sampling (negligible), letting the evaluated model solve the $\beta$ questions (i.e., $\beta F_{\text{model}}$) and initial summarization (21.52 TFLOPs). Together, these operations amount to about $F_{\text{init}} = \beta F_{\text{model}} + 27.6$ TFLOPs in total.

**Iteration (main source of computational cost)**  At each evaluation step, AutoJudger follows its agent-driven workflow. We denote the computation cost from AutoJudger as $F_{\text{AJ}}$, which includes:

- *Candidate Retrieval*: filter questions based on current ability estimates and ensure diversity.
- *Question Selection*: the AutoJudger agent analyzes retrieved candidates, incorporating dynamic memory and IRT-based model ability estimates to pick the most informative next question.

Table 13: Performance of AutoJudger with different IRT model configurations.

| Method | MMMU$_{DEV\&VAL}$ | MMT-Bench |
|---|---|---|
| Baseline (best) | 84.26 | 88.24 |
| 1PL | **87.94** | **92.06** |
| 2PL | 84.56 | 91.18 |
| 3PL | 86.03 | 91.91 |

Table 14: **Performance of different methods on MathVista-MINI.**

| Method | MathVista-MINI |
|---|---|
| Random | 85.44 |
| Cluster | 86.32 |
| Stratified | 85.15 |
| IRT | 83.09 |
| AutoJudger | **88.24** |

- *Memory Update*: update the memory table to track semantic coverage and difficulty distribution.

Combined with the computation cost of a single *Model Forward Pass* from the evaluated model, denoted as $F_{\text{model}}$, the total per-step iteration computation cost is

$$F_{\text{step}} = F_{\text{model}} + F_{\text{AJ}},$$

Using the MMT benchmark as an example, the computation cost of per-step candidate retrieval (<3 MFLOPs), per-step question selection (19.63 TFLOPs) and per-step memory update (3.62 TFLOPs), totaling $F_{\text{AJ}} = 23.25$ TFLOPs and $F_{\text{step}} = F_{\text{model}} + 23.25$ TFLOPs.

To compare AutoJudger with full-scale evaluation, we define the relative cost ratio as

$$R(\alpha, \beta, |Q|, F_{\text{model}}) = \frac{(\alpha|Q| - \beta) * F_{\text{step}} + F_{\text{init}}}{|Q| * F_{\text{model}}} = \alpha \cdot \frac{F_{\text{step}}}{F_{\text{model}}} + \frac{F_{\text{init}} - \beta * F_{\text{step}}}{|Q| F_{\text{model}}},$$

where $\alpha$ is the fraction of evaluation questions used, $\beta$ is the number of questions during initialization and $|Q|$ is the full size of evaluation benchmark. This formula means that, in practice, the relative cost ratio can be conservatively estimated by the fraction $\alpha$ of evaluation questions used and the per-step overhead of AutoJudger relative to the evaluated model's forward cost. Therefore, the computational overhead introduced by AutoJudger scales linearly with $\alpha$ and is bounded above by

$$R(\alpha, \beta, |Q|, F_{\text{model}}) \leq \alpha \cdot \frac{F_{\text{step}}}{F_{\text{model}}}.$$

This indicates that AutoJudger achieves significant computational savings compared to full-scale evaluation: by adaptively selecting only a small fraction of questions ($\alpha \ll 1$), the overall evaluation cost can be reduced by an order of magnitude while maintaining reliable ranking consistency.

When we take MMT benchmark as an example, the relative compuation cost is computed as:

$$R(\alpha, \beta, |Q|, F_{\text{model}}) = \alpha \left(1 + \frac{23.25}{F_{\text{model}}}\right) + \frac{27.6 - \beta * 23.25}{|Q| F_{\text{model}}} \approx \alpha \left(1 + \frac{23.25}{F_{\text{model}}}\right)$$

# I PROMPT OF AUTOJUGDER

---

### Prompt 1: Category Identification for Initialization Stage

```
You are an expert educational AI assistant specializing in question
    classification. Your task is to analyze the provided questions
    and categorize them into meaningful subject/topic categories.

Task Overview:
You will analyze a set of practice questions (including both text
    and images) and classify each question into a meaningful
    category(Expect two or more question in the same category).
The output should be a JSON object mapping question IDs to their
    respective categories.
{
Question ID: # Question ID
Difficulty: # Difficulty
Content: # Content
# IAMAG
Options: # Options
...
}

Output Requirements:
- Return a JSON object with the following format:
{
  "<Question_ID_1>": "<Category_Name_1>",
  "<Question_ID_2>": "<Category_Name_2>",
  ...
}
- Keys are question IDs (index) from the input data.
- Values are descriptive category names that you assign.
- ONLY return the JSON object; do not include any other text or
    explanation.
```

---

### Prompt 2: Category Identification for Iteration Stage

```
You are an expert educational classifier. Analyze the question and
    determine its category.
{
Question ID: # Question ID
Difficulty: # Difficulty
Content: # Content
# IAMAG
Options: # Options
...
}

Task: Review the question above. Determine all applicable
    categories from the existing list: {# Category}, or include new
    categories if necessary.

Output Requirements:
- Return a JSON object with:
{"category": ["Existing or new category name(s)"]}
- List ALL relevant categories (minimum 1 item).
- Use EXACT names for existing categories.
- Include multiple entries if needed (e.g., mixed existing/new
    categories).
- Do NOT add explanations, only JSON.
```

---

### Prompt 3: Question Recommendend

```
You are an expert educational AI assistant. Your task is to select
    the most appropriate next question from the candidate pool based
     on:
1. The student's current ability (# ability) estimated by IRT.
2. The diversity of question categories in the history.
3. The match between question difficulty and student ability.
Prioritize questions that balance category diversity and difficulty
    alignment.

Statistics in history questions
{
# Memory
}
Candidate Question Pool:
{
Question ID: # Question ID
Difficulty: # Difficulty
Content: # Content
# IAMAG
Options: # Options
...
}
Available IDs: # List of Question ID

Output JSON format:
{
    "summary": "Summary the Statistics in history questions.Don't
        merely state the facts; instead, synthesize deeper, abstract,
         and even metaphysical patterns or principles.",
    "think": "Reasoning here",
    "question_index": "SELECTED_ID"
}
Only return the JSON object. DO NOT explain.
```

---

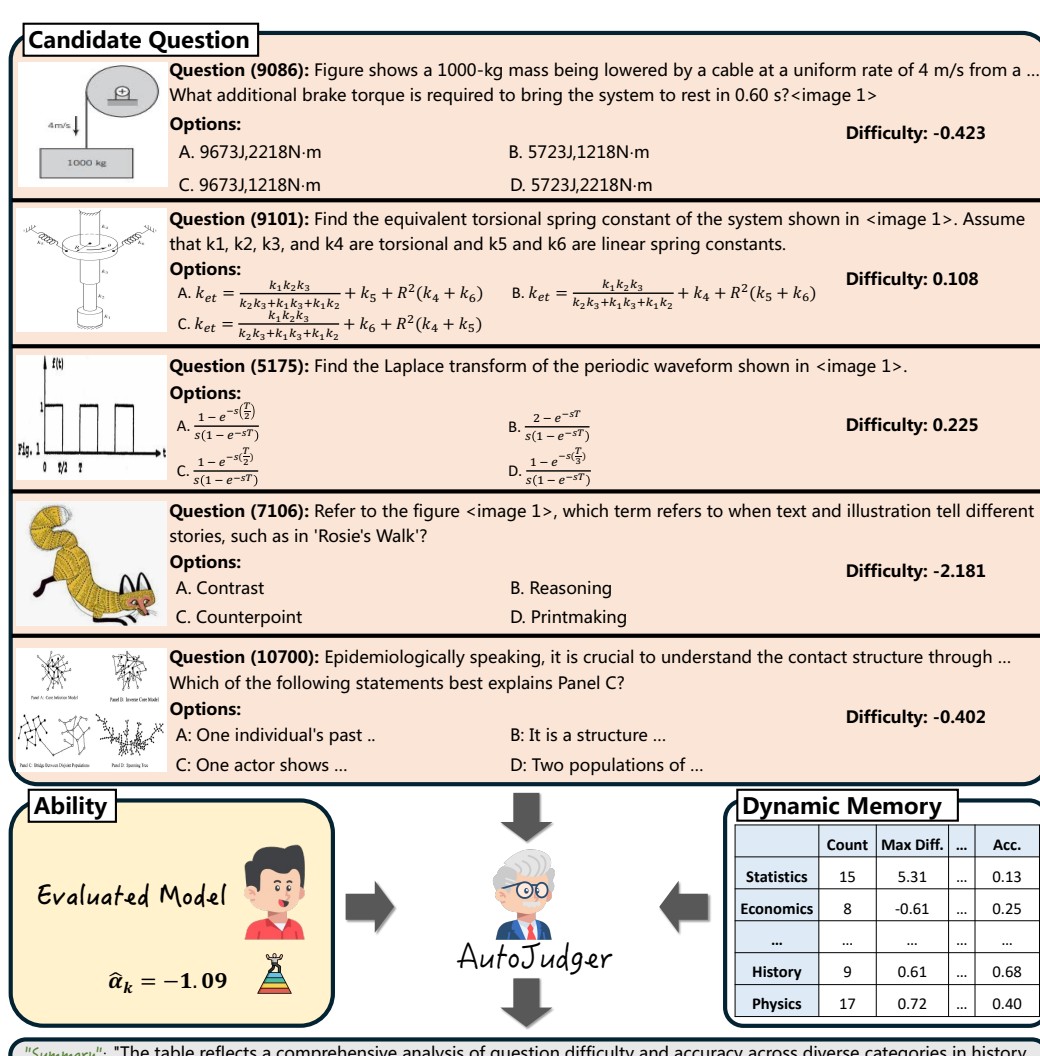

Figure 8: Response Examples from AutoJudger on MMMU.

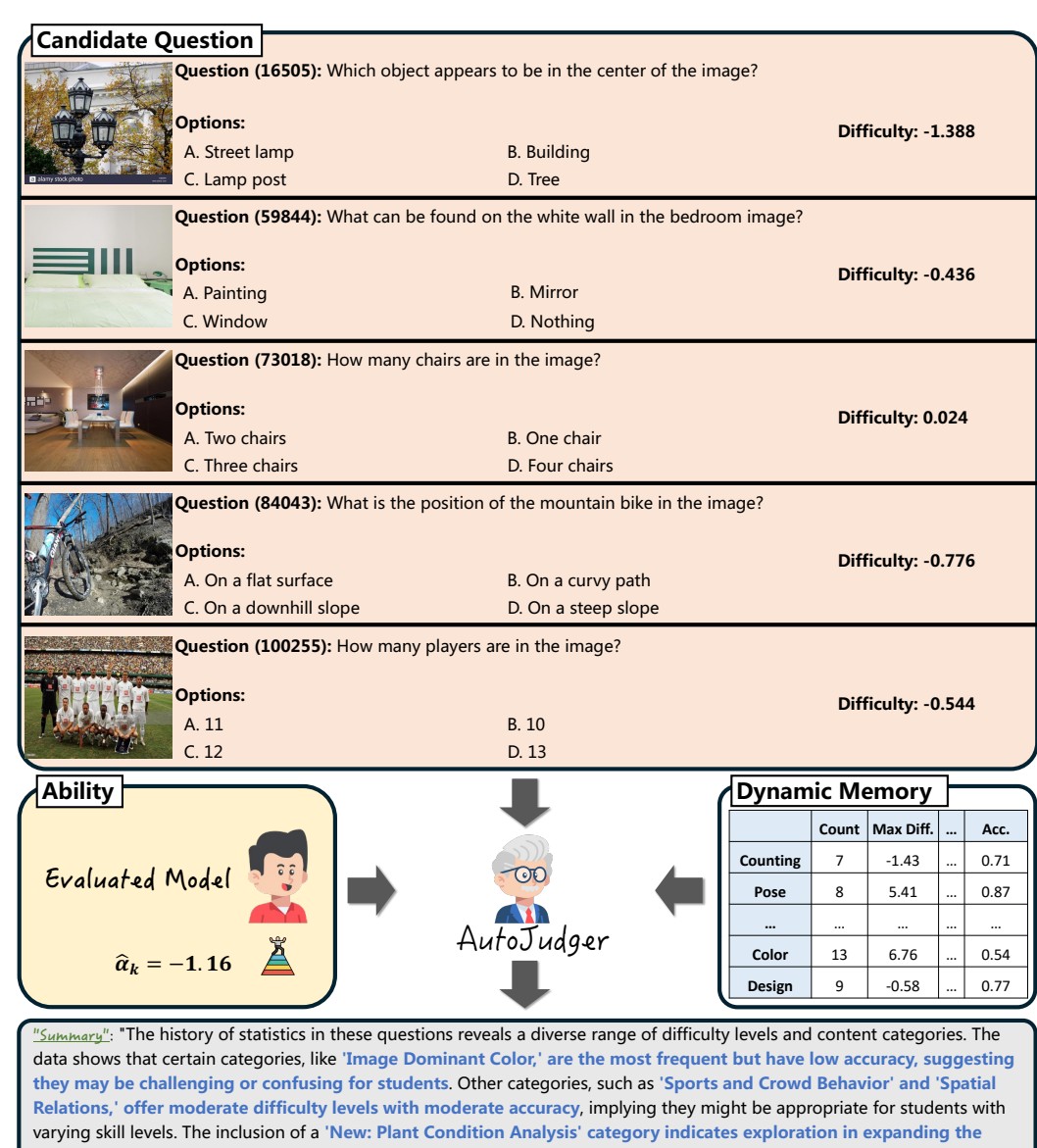

Figure 9: Response Examples from AutoJudger on SeedBench.

