# OpenReview forum: "AutoJudger: An Agent-Driven Framework for Efficient Benchmarking of MLLMs"
_ICLR.cc/2026/Conference — ICLR 2026 Conference Withdrawn Submission_

### Official Review · Reviewer_y3Fb · 2025-10-24

**Soundness:** 2
**Presentation:** 2
**Contribution:** 2
**Rating:** 4
**Confidence:** 3

**Summary:**

This work proposes an efficient benchmarking method for multimodal large language models (MLLMs), which leverages Item Response Theory (IRT) to estimate question difficulty and employs an autonomous evaluation agent to dynamically select the most informative test questions based on the model’s real-time performance.

**Strengths:**

1. The work is comprehensive, supported by extensive experiments validating the effectiveness of the proposed approach.
2. The paper focuses on efficient MLLM benchmarking, which is an important and practical problem for the community.

**Weaknesses:**

1. While the proposed framework is interesting, the paper lacks a deeper analysis of the fundamental factors contributing to its effectiveness. Beyond the ablation study, it remains unclear which core design choices are primarily responsible for the improvement. This makes it difficult to disentangle whether the observed benefits stem from `essential ideas` or the `complex agentic workflow`. It would be helpful to include a more prototype-level or simplified implementation as an additional baseline, to better isolate and discuss the key contributing factors, while reducing the emphasis on the agent workflow itself.
2. The comparison could be more comprehensive. If prior studies on efficient MLLM benchmarking exist, including them would make the contribution clearer and the improvement more convincing.

**Questions:**

1. I am curious whether there have been previous attempts or existing methods for efficient benchmarking of MLLMs？
2. The baselines in the paper appear relatively simple, while the proposed method introduces a much higher level of complexity. Could the authors clarify the necessity of such a complex design? Is it possible that a simpler approach might achieve similar effectiveness?

---

> ### Author Response · Authors · 2025-11-27
> **Response to Reviewer y3Fb**
>
> Dear Reviewer y3Fb,
>
> Thank you for your valuable feedback. Below are our responses to the concerns you raised.
>
> **Weakness 1 & Questions 2: Necessity of Core Component Design**
>
> > *W1: "While the proposed framework is interesting, the paper lacks a deeper analysis of the fundamental factors contributing to its effectiveness. Beyond the ablation study, it remains unclear which core design choices are primarily responsible for the improvement. This makes it difficult to disentangle whether the observed benefits stem from* *`essential ideas`* *or the* *`complex agentic workflow`**. It would be helpful to include a more prototype-level or simplified implementation as an additional baseline, to better isolate and discuss the key contributing factors, while reducing the emphasis on the agent workflow itself."*
>
> > *Q2*: *"The baselines in the paper appear relatively simple, while the proposed method introduces a much higher level of complexity. Could the authors clarify the necessity of such a complex design? Is it possible that a simpler approach might achieve similar effectiveness? "*
>
> **Response:**
>
> Please see our **general response to Common Question 1**. In short, the effectiveness of **AutoJudger** primarily stems from its three core components — ability dimension decomposition, ability state estimation, and adaptive selection — which are **model-agnostic and capture the key principles of active, information-driven evaluation**. While the agent workflow facilitates implementation, our ablation studies and prototype analyses confirm that rather than the complexity of the agent itself, **these fundamental design choices are the main contributors to the observed performance improvements**.
>
> ----
>
> **Weakness 2 & Questions 1: Comparison with Other Baseline Methods**
>
> > *W2: "The comparison could be more comprehensive. If prior studies on efficient MLLM benchmarking exist, including them would make the contribution clearer and the improvement more convincing."*
>
> > *Q1: "I am curious whether there have been previous attempts or existing methods for efficient benchmarking of MLLMs？"*
>
> **Response:**
>
> We appreciate the reviewer’s suggestion. In fact, we have thoroughly surveyed the literature on efficient multimodal large language model (MLLM) evaluation, and, to the best of our knowledge, **our work presents the first adaptive framework for MLLM benchmarking**. Therefore, our comparison encompasses the most representative prior methods used in LLMs [1-4], and a detailed description of these approaches is provided in Section 4.1 of the paper.
>
> To further investigate the design of AutoJudger, we also implemented several other baselines. Please refer to our **general resopnse to Common Question 1.4** for more details. In short, we find the agent-based instantiation of AutoJudger performs the best.
>
> [1] Yotam Perlitz, Elron Bandel, Ariel Gera, Ofir Arviv, Liat Ein-Dor, Eyal Shnarch, Noam Slonim, Michal Shmueli-Scheuer, and Leshem Choshen. Efficient benchmarking of language models. arXiv preprint arXiv:2308.11696, 2023. (This paper corresponds to "Stratified" baseline)
>
> [2] Alec Radford, Jong Wook Kim, Chris Hallacy, Aditya Ramesh, Gabriel Goh, Sandhini Agarwal, Girish Sastry, Amanda Askell, Pamela Mishkin, Jack Clark, et al. Learning transferable visual models from natural language supervision. In International conference on machine learning, pp. 8748–8763. PmLR, 2021. (This paper corresponds to "Cluster" baseline)
>
> [3] Frederic M Lord. Applications of item response theory to practical testing problems. Routledge, 2012. (This paper corresponds to "IRT" baseline)
>
> [4] Yan Zhuang, Junhao Yu, Qi Liu, Yuxuan Sun, Jiatong Li, Zhenya Huang, and Enhong Chen. Efficient Benchmarking via Bias-Bounded Subset Selection. IEEE Transactions on Pattern Analysis and Machine Intelligence, 2025. (This paper uses both "IRT" and "Stratified" baselines.)
>
> ---
>
> Thank you again for these valuable suggestions, which have greatly helped improve the quality of this paper. We also warmly welcome any further feedback and discussions.
>
> Authors

---

### Official Review · Reviewer_ipfW · 2025-10-30

**Soundness:** 2
**Presentation:** 2
**Contribution:** 2
**Rating:** 4
**Confidence:** 3

**Summary:**

This paper presents AutoJudger, an agent-driven framework designed to address the escalating computational cost of benchmarking MLLMs. The core methodology leverages Item Response Theory (IRT) to pre-estimate question difficulty. This information is then used by an autonomous agent to adaptively select a subset of informative questions based on the real-time performance of the model under evaluation. The framework incorporates a semantic-aware retrieval mechanism to ensure question diversity and a dynamic memory module to maintain contextual statistics, guiding a coherent and globally informed selection process.
The authors report that this adaptive approach significantly reduces evaluation costs, achieving over 90% ranking consistency with the full-benchmark results while using as little as 4% of the data on benchmarks like MMT-Bench. The work's primary contribution is a shift from static, fixed-subset evaluation to a dynamic, model-specific testing protocol that aims to balance efficiency with reliability.

**Strengths:**

- The paper's primary originality lies in the principled reframing of MLLM benchmarking from static subset sampling to a dynamic, agent-driven "interview" process. This is a significant conceptual shift. The creative synthesis of established concepts from different fields—Item Response Theory (IRT) from psychometrics, semantic-aware retrieval, and a dynamic memory module—into a cohesive framework to solve this problem is highly novel and insightful.

- The paper is exceptionally well-written and clearly structured. The core "interviewer" agent paradigm is an intuitive and effective metaphor that guides the reader through the complex process. Figure 2 provides an excellent, comprehensive overview of the entire dynamic evaluation loop, making the complex interactions between components easy to understand and follow.

**Weaknesses:**

- The framework's effectiveness is heavily reliant on the capability of the "interviewer" agent (Qwen2.5-VL-7B). This conflates the performance of the framework's mechanics (IRT, memory) with the reasoning power of a specific, strong MLLM. A weaker agent might make suboptimal choices, and the agent's inherent biases could lead to systematically skewed question selections.

- The premise of using IRT requires pre-estimating question difficulties, which the authors accomplish by collecting responses from 60 offline models on the full benchmarks. This represents a substantial, and for many researchers or new benchmarks, prohibitive, upfront computational barrier. While amortized, it limits the framework's practical applicability to new or evolving datasets.

**Questions:**

Plz answer my concerns in the weakness section

---

> ### Author Response · Authors · 2025-11-27
> **Response to Reviewer ipfW**
>
> Dear Reviewer ipfW,
>
> Thank you for your constructive suggestions. Below are our responses to the concerns you raised.
>
> **Weakness 1：Validity and Generalizability of the AutoJudger Framework**
>
> > *"The framework's effectiveness is heavily reliant on the capability of the "interviewer" agent (Qwen2.5-VL-7B). This conflates the performance of the framework's mechanics (IRT, memory) with the reasoning power of a specific, strong MLLM. A weaker agent might make suboptimal choices, and the agent's inherent biases could lead to systematically skewed question selections."*
>
> **Weakness 1.1: Justification for the AutoJudger Design:**
>
> **Response:**
>
> We provide further discussion of the AutoJudger design justification in our **general response to Common Question 1**.  In short, AutoJudger is inspired by human cognitive processes: it treats evaluation as an active, adaptive inference task rather than passive scoring. The framework maintains estimates of the model’s abilities, updates them based on past responses, and adaptively selects questions that are maximally informative across multiple dimensions. AutoJudger operationalizes this perspective through three core components—ability decomposition, ability estimation, and adaptive selection—allowing efficient and fine-grained assessment of MLLM capabilities.
>
> **Weakness 1.2**: **Potential Bias in Question Selection stemming from inherent biases within the AutoJudger agent's base model.**
>
> **Response:**
>
> - **Clarification of the Question Selection Task:** First, we would like to clarify that **the task of AutoJudger is to select appropriate questions based on current information**—including the semantics and difficulty of candidate questions, and the current capability level of the model under test—**rather than to answer the questions itself** (where noticeable discrepancies exist between different models). The difference between these two tasks means that the bias observed in the MLLM on the latter task is unlikely to transfer to the former.
> - **Empirical Validation with Different base models:** In Appendix E.3, we implement AutoJudger with different MLLMs as the base model. The results show that AutoJudger (1) consistently outperforms all baseline methods under different backbones, and (2) maintains stable performance even when using smaller-sized backbones. These results provide strong evidence that **AutoJudger is robust to model variation and that potential foundation model bias does not materially affect the evaluation outcome**.
>
> ---
>
> **Weakness 2：Justification of IRT-Based Difficulty Assessment and Feasible Alternatives**
>
> > *"The premise of using IRT requires pre-estimating question difficulties, which the authors accomplish by collecting responses from 60 offline models on the full benchmarks. This represents a substantial, and for many researchers or new benchmarks, prohibitive, upfront computational barrier. While amortized, it limits the framework's practical applicability to new or evolving datasets."*
>
> **Response:**
>
> Please see our **general response to Common Question 3**. In short, we have demonstrated that under our framework, **three resource-constrained alternatives still yield strong performance**. Across all these settings, **our method consistently performs the best**, showing that it remains effective and robust.
>
> ---
>
> Thank you again for these valuable suggestions, which have greatly helped improve the quality of this paper. We also warmly welcome any further feedback and discussions.
>
> Authors

---

### Official Review · Reviewer_9X6p · 2025-11-01

**Soundness:** 2
**Presentation:** 3
**Contribution:** 3
**Rating:** 4
**Confidence:** 3

**Summary:**

This paper introduces AutoJudger, an agent-driven framework designed to address the escalating cost of evaluating MLLMs. The core method uses Item Response Theory to pre-calculate a difficulty score for all benchmark questions and to estimate a model's ability in real-time during the evaluation. An MLLM-based agent acts as an "interviewer," adaptively selecting the most informative questions by analyzing the model's current ability, the difficulty of the questions, and a dynamic memory that tracks statistical coverage across different topics. The framework was evaluated on four representative multimodal benchmarks, including MMMU, MMT-Bench, and SEEDBench, demonstrating that it can maintain over 90% ranking accuracy with the full benchmark results while using as little as ~5% of the data.

**Strengths:**

- The paper proposes a novel, agent-driven framework to address the practical problem of expensive MLLM evaluation costs. The AutoJudger framework is built on a principled and well-suited foundation, adapting Item Response Theory to define the problem difficulty and dynamically estimate the model's ability.

- The design and implementation of the framework are solid and reasonable. The core components—real-time ability estimation, semantic-aware retrieval, an agent-based selection module, and a statistical dynamic memory—work together logically to achieve efficient and adaptive evaluation.

- The method demonstrates good empirical results, showing it can maintain high ranking accuracy (e.g., >90% on MMT-Bench) while using a small fraction of the original data (e.g., 4%). The robustness of the framework is well-supported by ablation studies, including testing different agent backbones.

**Weaknesses:**

**The most important issue:** The primary metric, "Ranking Accuracy" (Sec 4.1), seems to be a significant limitation in the evaluation. This ordinal metric only tells if the relative order of models is preserved and does not measure if the cardinal score gaps are retained. A framework that shrinks a 20-point performance gap (on the full benchmark) to a 1-point gap would still achieve 100% ranking accuracy, but users won't trust such a framework even if it's much more efficient and cheap.

Also, the framework's practicality is questionable when evaluating smaller models. The agent itself is a medium-sized MLLM (e.g., 7B), introducing a substantial, fixed computational overhead per step. As shown in Table 4, this overhead is proportionally massive (e.g., 4.85x per step for a 3B model), which undermines the goal of cost-saving for this common use case (Consider that many researchers in academia do experiments with small models).

**Questions:**

The paper takes a valuable direction and proposes an interesting automatic evaluation framework for MLLMs. However, the evaluation only discusses how the ranking can be preserved without measuring whether the score gap can also be retained under reduced evaluation costs, making the conclusion less convincing.

---

> ### Author Response · Authors · 2025-11-27
> **Response to Reviewer 9X6p**
>
> Dear Reviewer 9X6p,
>
> Tahnk you for your detailed advices. Below are our responses to the concerns you raised.
>
> **Weakness 1 & Question 1: Additional metrics that can preserve the absolute score gaps.**
>
> > *"**The most important issue:** The primary metric, "Ranking Accuracy" (Sec 4.1), seems to be a significant limitation in the evaluation. This ordinal metric only tells if the relative order of models is preserved and does not measure if the cardinal score gaps are retained. A framework that shrinks a 20-point performance gap (on the full benchmark) to a 1-point gap would still achieve 100% ranking accuracy, but users won't trust such a framework even if it's much more efficient and cheap."*
>
> **Response:**
>
> Thank you for raising this insightful and important concern.
>
> To directly address this point, we conducted an additional analysis to evaluate whether the **absolute gaps** between estimated model abilities from AutoJudger (using only 5% of evaluation data) are meaningfully aligned with the **true accuracy gaps** observed on the full benchmark. Specifically, for all pairs among the 17 evaluated MLLMs, we computed:
>
> - The difference in IRT-estimated abilities using 5% sampled questions (our proxy scores), and
> - The difference in actual accuracy on the full benchmark (ground-truth scores).
>
> We then fit a zero-intercept linear regression model to these paired differences (i.e., $\Delta \theta_{\text{5\%}} \rightarrow \Delta \text{Acc}_{\text{full}}$) and report the Pearson correlation coefficient $r$ as a measure of linear correspondence. As shown in the table below, **the average correlation across the three datasets exceeds 0.90,** indicating a **strong linear relationship** between the estimated ability gaps and the true accuracy gaps—even under extreme subsampling.
>
> ***Rebuttal Table 6: Correlation between the difference of IRT-estimated abilities and actual accuracy.*** Compared with other baselines, AutoJudger yields the best correlation with true accuracy gaps observed on the full benchmark.
>
> | Dataset       | correlation r (zero-intercept fit) |        |       |
> | ------------- | ---------------------------------- | ------ | ----- |
> |               | AutoJudger                         | Random | IRT   |
> | AI2D_TEST     | **0.981**                          | 0.977  | 0.941 |
> | MMMU_DEV_VAL  | **0.850**                          | 0.831  | 0.784 |
> | MMT-Bench_VAL | **0.924**                          | 0.861  | 0.894 |
> | **Average**   | **0.918**                          | 0.890  | 0.873 |
>
> Overall, these results strongly suggest that our framework does **not only preserve relative ranking order, but also faithfully approximates the absolute performance margins between models.**
>
> -----
>
> **Weakness 2: Computational Overhead on Smaller Models**
>
> > *"Also, the framework's practicality is questionable when evaluating smaller models. The agent itself is a medium-sized MLLM (e.g., 7B), introducing a substantial, fixed computational overhead per step. As shown in Table 4, this overhead is proportionally massive (e.g., 4.85x per step for a 3B model), which undermines the goal of cost-saving for this common use case (Consider that many researchers in academia do experiments with small models)."*
>
> **Response2:**
>
> Please see our **general response to Common Question 2**. In short, even when using a medium-sized AutoJudger (e.g., 7B) to evaluate smaller models, **the overall cost remains substantially lower than standard full-benchmark evaluation**. As discussed, AutoJudger only scores a small subset of samples, and using a smaller AutoJudger (e.g., 3B) further reduces computational overhead while maintaining comparable or even better performance in some benchmarks. This demonstrates that **AutoJudger** **remains practical and cost-efficient for common academic scenarios involving smaller models**.
>
> ---
>
> Thank you again for these valuable suggestions, which have greatly helped improve the quality of this paper. We also warmly welcome any further feedback and discussions.
>
> Authors

---

### Official Review · Reviewer_UV1R · 2025-11-02

**Soundness:** 2
**Presentation:** 3
**Contribution:** 1
**Rating:** 4
**Confidence:** 4

**Summary:**

This paper proposes an autonomous MLLM evaluation pipeline that dynamically selects the most informative questions from the question pool, with the aid of agents. In the core of the pipeline, it measures the difficulty (or score) of a question using Item Response Theory based on the model's response, subsequently it selects a small set of questions from the entire benchmarks. Finally, the selected question subset is viewed as a budget-friendly evaluation benchmark proxy.

Trimming down the evaluation benchmark to lower the cost is indeed an interesting idea. Nevertheless, the cost saved by trimmed benchmarks seems trivial compared with the massive cost of training or post-training.

**Strengths:**

1. Nicely presented paper, easy to follow, and well organized. The problem is interesting; the evaluation cost of MLLMs indeed often gets ignored.
2. The AutoJudger framework seems reasonable to me, and the idea of utilizing an autonomous framework to mine difficult questions for MLLM evaluation is worth promoting, especially since some recent works have pointed out that some benchmarks may suffer from data leakage or lower-quality question issues.

**Weaknesses:**

1. The need to trim down benchmark size **needs more solid evidence.** For example, the CO2 emission on a full benchmark, or a more straightforward measure such as GPU rental cost?
   * I know section 4.4 has already mentioned the cost, but it will be nice to see a number in $.
   * As mentioned in the summary, the evaluation cost can be trivial compared with the massive cost of training (including post-training/fine-tuning) of a model.
2. It is good to see that the author provides how different methods perform under different compression ratios; however, **it is unclear why the compression ratio is capped to 5%.** For example, 10% or 15% compression is a good step up compared with utilizing the full benchmark already.
   * I consider this a weakness because I doubt that randomly selecting 10% or maybe 15% questions from a benchmark can yield a satisfying result already. I strongly recommend that the author clarify why it is capped at 5% and how it behaves if the number is set to a higher value, say 15% or 10%.
3. The IRT-based difficulty assessment requires offline evaluation results of different models. I wonder if this will compromise the contribution of the proposed method. In fact, I respectfully **disagree** with the claim that the cost of question difficulty estimation is **negligible** simply because it is precomputed (Section H in the appendix), especially since it involves evaluating 60 models on full benchmarks.
   * Also, will the evaluation result on outdated models be useful for the difficulty assessment?

**Questions:**

see weakness

---

> ### Author Response · Authors · 2025-11-27
> **Response to Reviewer UV1R**
>
> Dear Reviewer UV1R,
>
> We are truly grateful for your detailed comments and constructive feedback. Below are our responses to the concerns you raised.
>
> **Weakness 1: Practical Cost Reduction of AutoJudger**
>
> **Response:**
>
> We appreciate the reviewer’s concern regarding the practical evaluation cost. A detailed analysis of computational efficiency and monetary cost has been provided in our **general** **response to Common Question** **2**, where we report per-sample GPU time, total cost on a commercial cloud platform, and the resulting 8×–20× overall savings.
>
> In addition, referring to the evaluation settings used in the Qwen2.5-VL technical report, we note that **most mainstream benchmarks are at least of 1k sample scale**. In our experiments, **we have already covered representative benchmark sizes**—from small (1k, MMMU_VAL) to medium (3k, MMT-Bench_VAL) to large (14k, SeedBench-IMG). Therefore, **AutoJudger demonstrates strong efficiency across both small and large evaluation benchmarks**.
>
> In practice, the researchers training the model are often not the same as the researchers evaluating it. Academic groups, community leaders, or third-party organizations frequently perform independent evaluations. For them, the evaluation burden is not “trivial” compared to pretraining — it is the primary cost they must bear. Even for companies training proprietary models, internal evaluation teams operate under separate budgets and must repeatedly test each checkpoint, making evaluation a significant operational overhead.
>
> -----
>
> **Weakness 2: Clarification and Extension of Compression Ratio Choices**
>
>
> **Response:**
>
> We thank the reviewer for the suggestion regarding compression ratios. In the original submission, we selected 5% to test the framework under an extreme low-sample setting. To address the concern, we conducted additional experiments with 10% and 15% compression. The results are summarized below:
>
> ***Rebuttal Table 5: Performance of different methods under varying compression ratios.***
>
> | Method     | MMT-Bench_VAL |       |       | \|   | AI2D_TEST |       |       | \|   | MMMU_DEV_VAL |       |       |
> | ---------- | ------------- | ----- | ----- | ---- | --------- | ----- | ----- | ---- | ------------ | ----- | ----- |
> |            | 5%            | 10%   | 15%   | \|   | 5%        | 10%   | 15%   | \|   | 5%           | 10%   | 15%   |
> | Random     | 85.88         | 90.88 | 94.56 | \|   | 93.82     | 95.74 | 96.62 | \|   | 81.47        | 83.68 | 88.38 |
> | Cluster    | 87.79         | 93.09 | 93.24 | \|   | 93.53     | 94.71 | 96.32 | \|   | 78.97        | 83.53 | 88.53 |
> | Stratified | 84.12         | 90.29 | 90.59 | \|   | 93.97     | 94.41 | 94.71 | \|   | 84.26        | 87.06 | 87.65 |
> | IRT        | 88.24         | 90.44 | 91.91 | \|   | 89.71     | 91.91 | 93.38 | \|   | 82.35        | 86.76 | 90.44 |
> | AutoJudger | 94.85         | 98.53 | 99.26 | \|   | 96.32     | 97.06 | 97.06 | \|   | 87.5         | 91.18 | 91.18 |
>
>
> The results indicate that **AutoJudger consistently outperforms other methods across all datasets**, and its performance steadily improves as the compression ratio increases, approaching the full benchmark results. Under the same data budget, AutoJudger outperforms all the other baselines by a clear margin.
>
> Furthermore, we also consider the comparison between different compression ratios:
>
> **Comparison between different methods with similar total costs** For ease of comparison, we assume all evaluated models are 7B. Notably, the cost of evaluating 10% of questions at random is roughly comparable to that of AutoJudger selecting 5% of questions. Under this comparable budget, AutoJudger achieves significantly higher performance than the 10% random strategy.
>
> **Comparison between different methods with similar performance** Similarly, when comparing 15% random selection with AutoJudger selecting 5% of questions, the performance is roughly comparable; however, the computational cost of the 15% random strategy is approximately 1.65× that of AutoJudger (5%).
>
> These findings demonstrate that **AutoJudger is both robust in few-shot scenarios and scalable to higher compression ratios, effectively balancing evaluation efficiency and accuracy**.
>
> -----
>
> **Weakness 3: Justification of IRT-Based Difficulty Assessment and Feasible Alternatives**
>
> **Response:**
>
> Please see our response to **Common Question 3**. In short, we have demonstrated that **under our framework, three resource-constrained alternatives still yield strong performance.** Across all these settings, **our method consistently performs the best**, showing that it remains **effective** and **robust**.
>
> -----
>
> Thank you again for these valuable suggestions, which have greatly helped improve the quality of this paper. We also warmly welcome any further feedback and discussions.
>
> Authors

---

### Author Response · Authors · 2025-11-27
**General Response to Common Questions (Part 3 of 3)**

**Common Question 3: Cost of question difficulty Measurement**

**Response:**

We fully agree that minimizing computational overhead is central to our framework’s design goals. At first glance, the strategy of collecting responses from 60 offline models to pre-estimate question difficulties may appear costly. However, in practice, this cost is often negligible: most modern models are already evaluated on major benchmarks at release time, and conversely, **new benchmarks routinely include evaluations across a suite of established models**. As a result, the necessary historical response data are typically already available in the literature or public leaderboards — effectively "free" in terms of additional computation (Even though the number might be fewer than 60, it satisfies the requirements of Alternative 2, as described below, and its performance is assured). Moreover, for the fully explored benchmarks we consider, using 60 models provides sufficient coverage to accurately estimate question difficulties without imposing excessive overhead.

Nonetheless, we acknowledge that **this assumption may not always hold**, especially for very new tasks, proprietary datasets, or domains where public evaluation logs are scarce. **In Appendix E.2 (Page 20, Line 1074 -- 1079), we have discussed most resource-constrained circumstances where no prior historical responses are available**, thus we provide the alternative (**Alternative 1**) by assigning a default difficulty value (e.g. 0) to all questions.

To address more practical scenarios where historical data is limited, we **explored two extra alternatives**:

- When a benchmark is released with ***limited historical responses*** from evaluated models, we can choose **Alternative 2: Estimating difficulties from a much smaller set of historical model responses** (e.g., responses from only the 20 most recently released models).
- When we are faced with a brand new benchmark and we have a capable LLM that can serve as a proxy for difficulty labeling, we can choose **Alternative 3: Using a commercial large language model API (e.g., GPT-4o) to directly annotate the difficulty.** In practice, we prompt the model to assign each sample a **five-level difficulty score** drawn from {-1, −0.5, 0, 0.5, 1}.

We compare these alternatives with the original strategy used in our paper (namely "original"). Results are summarized in the table below. While our original strategy does yield best performance gains, other alternatives still lead to competitive results — consistently outperforming the best baseline by a clear margin. This demonstrates that **our framework remains effective and robust across a spectrum of practical conditions**, including those where high-quality difficulty measurement is not feasible.

***Rebuttal Table 4: Performance of AutoJudger under different question difficulty measurement***

| Method           | MMMU_DEV_VAL | MMT-Bench_VAL |
| ---------------- | ------------ | ------------- |
| Baseline(best)   | 84.26        | 88.24         |
| AutoJudger       |              |               |
| -- Alternative 1 | 86.76        | 89.71         |
| -- Alternative 2 | 86.76        | 91.18         |
| -- Alternative 3 | 85.29        | 88.97         |
| -- Original      | 87.94        | 92.06         |

---

### Author Response · Authors · 2025-11-27
**General Response to Common Questions (Part 2 of 3)**

**Common Question 2: The practical cost of AutoJudger.**

**Response:**

In Section 4.4 and  Appendix H, our analysis of computational cost focused on theoretical relative overhead, measured by FLOPs.  To provide a more intuitive comparison, we use MMT-Bench_VAL as a concrete case study and report both **compute cost (GPU seconds per sample)** and **total monetary cost** under the same hardware and software environment. All experiments were conducted on RunPod (RTX4090 at $0.59 per GPU hour). The Qwen2.5-VL-72B model uses 8×4090 GPUs, while all other models use a single GPU.

***Rebuttal Table 2: The practical cost of AutoJudger.***

| Evaluation Method                           | Evaluated Model   |                   |                    |
| ------------------------------------------- | ----------------- | ----------------- | ------------------ |
|                                             | **Qwen2.5-VL-3B** | **Qwen2.5-VL-7B** | **Qwen2.5-VL-72B** |
| ***Compute Cost (GPU Seconds per Sample)*** |                   |                   |                    |
| Standard Evaluation                         | 4.52              | 6.83              | 781.6              |
| AutoJudger(7B)                              | 10.11             | 12.42             | 787.19             |
| AutoJudger(3B)                              | 8.18              | 10.49             | 785.26             |
| ***Total Cost (USD)***                      |                   |                   |                    |
| Standard Evaluation                         | 2.32              | 3.5               | 400.96             |
| AutoJudger(7B)                              | 0.26              | 0.32              | 20.15              |
| Cost Reduction(7B)                          | 8.92×             | 10.94×            | 19.90×             |
| AutoJudger(3B)                              | 0.21              | 0.27              | 20.1               |
| Cost Reduction(3B)                          | 11.05x            | 12.97x            | 19.95x             |

***Rebuttal Table 3: Performance of AutoJudger Across different backbone sizes.***

| Method         | AI2D_TEST | MMMU_DEV_VAL | MMT-Bench_VAL |
| -------------- | --------- | ------------ | ------------- |
| Baseline(best) | 93.97     | 84.26        | 88.24         |
| AutoJudger(3B) | 97.06     | 88.97        | 89.71         |
| AutoJudger(7B) | 94.85     | 87.94        | 92.06         |

Through these results, we observe two key findings as shown in Rebuttal Table 2 and 3.

1. Although AutoJudger introduces additional computation due to the selection model, **the overhead of running our framework remains modest and well-bounded**. For example, the per-sample cost increases from 4.52 to 8.18 seconds — an increase in absolute terms, but still far smaller than the cost of evaluating large models such as 72B, where a single forward pass dominates the total evaluation budget. In other words, the extra overhead is not the bottleneck in the evaluation pipeline.
2. The total evaluation cost in Standard Evaluation mainly comes from running the *target model* over the *entire benchmark*. AutoJudger substantially reduces this cost by only scoring a small subset of samples. Under our default setting (evaluating 5% of the benchmark), **the total monetary cost is reduced by nearly 10× across all model sizes, and up to 20× for the largest model.** For instance, on a 1k-sample benchmark, evaluating a 7B model costs about 1.12 under standard evaluation, whereas AutoJudger(3B) reduces this to only 0.06 — saving approximately 1.06, which corresponds to a 94.6% decrease in cost for a single evaluation run. Importantly, using a 3B AutoJudger does not significantly compromise performance, and in some benchmarks even outperforms the 7B AutoJudger, demonstrating that problem-solving ability does not necessarily equate to effective selection ability.

Overall, **AutoJudger preserves evaluation quality while achieving order-of-magnitude cost savings**, making it highly practical for benchmarking in real-world scenarios.

---

### Author Response · Authors · 2025-11-27
**General Response to Common Questions (Part 1 of 3)**

We thank all the reviewers for their valuable comments and constructive suggestions. Below we summarize and response to the common questions raised by the reviewers.

**Common Question 1: The deep analysis and justification for the design of AutoJudger.**

**Common Question 1.1: Clarification on the motivation behind AutoJudger.**

**Response:**

Cognitive psychology suggests that we never observe “ability” directly: we only see behavior and form hypotheses about latent mental traits [1–2]. Humans routinely do this through **active, adaptive querying**—asking questions, posing tasks, and refining their internal estimation of another person based on the answers. A job interview is an example of this process. The interviewer starts with a few broad, relatively simple questions, then selectively increases difficulty along promising skill dimensions or shifts to new dimensions when a particular area appears saturated or clearly weak. Under strict time and question budgets, the interviewer’s goal is not to exhaustively test every skill, but to **efficiently infer a multi-dimensional ability profile and its boundaries**.

We take this perspective as a blueprint for evaluating modern MLLMs, and thereby propose our framework **AutoJudger**. Unlike traditional evaluations that passively score a fixed set of questions, AutoJudger treats the evaluation process as an **active cognitive process:** it maintains a set of hypothesized ability dimensions, continuously updates beliefs about the model competence based on prior answers, and adaptively selects evaluation items that are maximally informative.

---

**Common Question 1.2: General framework definition for AutoJudger.**

**Response:**

Formally, we formalize AutoJudger as a generic adaptive evaluation framework and implement it through three core components:

1. **Ability Dimension Decomposition:** Define the ability dimensions to evaluate — either from explicit dataset labels/taxonomies or via implicit semantic patterns inferred from evaluated questions — and associate each question with one or more dimensions.
2. **Ability State Estimation:** Maintain an updated estimate of the model’s abilities based on historical responses, by using heuristic methods, IRT or other cognitive diagnostic models.
3. **Adaptive** **Question** **Selection:** Given current ability estimates and the question pool, select the next question to probe near ability boundaries while preserving coverage over different semantic regions.

By design, this framework has several practical applications. It **provides a principled way for efficient benchmarking**: it dynamically adjusts evaluation content based on evolving understanding of the model, selects a small subset of questions that can nearly recover full-benchmark ranking, and preserves ability differences. It supports **targeted, skill-aware diagnostics** along **decomposed ability dimensions,** enabling flexible differentiation across disciplines (e.g., mathematical reasoning, code generation, scientific understanding), modalities (e.g., pure text, vision–language, tables and structured data), and task formats (e.g., factual question answering, open-ended generation, tool use, and planning).

We summarize the AutoJudger framework in **Algorithm 1.**

```C++
Input:
    D           // Question pool with metadata (taxonomy, semantic embedding, difficulty, etc.)
    B           // Evaluation budget (max number of questions or queries)
Output:
    θ           // Ability profile of the evaluated model

// 1. Initialize ability dimensions and memory
DIMENSIONS ← INIT_ABILITY_DECOMPOSITION(D)
θ ← INITIAL_ABILITY_ESTIMATE(DIMENSIONS)

// 2. Adaptive evaluation loop
for t = 1 to B do
    // 2.1 Select the next question based on current ability estimates and coverage
    x_t ← QUESTION_SELECT_STRATEGY(D, DIMENSIONS, θ)

    // 2.2 Query the evaluated model and observe response
    y_t ← QUERY_EVALUATED_MODEL(x_t)

    // 2.3 Update ability decomposition and estimates
    DIMENSIONS ← UPDATE_ABILITY_DECOMPOSITION(DIMENSIONS, x_t, y_t)
    θ ← ESTIMATE_ABILITY(θ, DIMENSIONS)

    // 2.4 (Optional) Early stopping
    if CHECK_STOP(θ, DIMENSIONS, t, B) then
        break
    end if
end for

// 3. Return final ability assessment
return θ
```

It is worth emphasizing that AutoJudger is a **general framework** rather than a single fixed algorithm: each of the three components above allows multiple, interchangeable implementations. Building on this, we further reorganize the main results of this paper based on different implementation details of the three core components (detailed in our response to Common Question 1.4).

---

> ### Author Response · Authors · 2025-11-27
> **General Response to Common Questions (Part 1 of 3)--Supplementary Note 1**
>
> **Common Question 1.3: Reasons behind the agent-based instantiation of AutoJudger.**
>
> **Response**:
>
> The key motivation to build the AutoJudger framework upon an **agentic workflow** stems from the target to realize a generalizable evaluation system capable of engaging with and assessing diverse, complex benchmark environments.
>
> Therefore, analogous to a human evaluator, we instantiate our AutoJudger agent with a “brain” powered by an MLLM. The advantages of this approach are twofold:
>
> 1. **Generalizability to Benchmark Structure:** Our system can directly utilize the **predefined capability taxonomy** of certain benchmarks. When such information is missing, it can leverage our designed **memory mechanism** during the dynamic evaluation process to gradually construct a capability analysis system that supports evaluation by utilizing **implicit semantics** within the questions.
> 2. **Dynamic Integration of Capability Dimensions and Difficulty:** To accurately probe the boundaries of capabilities with a limited amount of evaluation data, the difficulty of the selected questions must match the model's current ability. This provides the most effective **information gain** (i.e., routing strong performance to more difficult questions and weak performance to easier ones). By integrating an **IRT model**, the dynamic nature of our agent framework precisely supports the **joint** **consideration** of both the capability dimension (semantics) and question difficulty during the selection process.
>
> To substantiate these design choices, we have conducted **a series of ablation studies shown in Table 2 of our paper.** We would like to **highlight the results of "AutoJudger w/o agent"**, where we (1) replace the semantic dimensions summarized by the agent with embedding-based semantic clusters, and (2) use a purely difficulty-based sampler which does not support semantic reasoning like agent. This variant of AutoJudger can also been taken as a variant of our "Cluster" baseline, as we have summarized in the table below. As shown, the results confirm the significant drop of ranking accuracy on the challenging benchmarks like MMMU and MMT-Bench, validating that **the simple IRT-guided difficulty sampling is insufficient on its own**. For a more rigorous comparison, we introduce a hybrid question-selection strategy (**Hybrid**) on top of the AutoJudger w/o agent variant, which allows the selection process to equally account for both question difficulty and semantic distance (implementation details are in our response to Common Question 1.4). Taken together, these ablations demonstrate that **our agent-based instantiation is reasonable and effective.**
>
> | Method               | Ability Decomposition     | Ability Estimation | Question Selection    | MMMU  | MMT-Bench |
> | -------------------- | ------------------------- | ------------------ | --------------------- | ----- | --------- |
> | Cluster              | embedding-based semantics | Accuracy           | Semantic              | 78.97 | 87.79     |
> | AutoJudger w/o agent | embedding-based semantics | IRT                | Difficulty            | 82.87 | 86.62     |
> | Hybrid               | embedding-based semantics | IRT                | Difficulty + Semantic | 85.29 | 89.71     |
> | AutoJudger           | agent-based semantics     | IRT                | Difficulty + Semantic | 87.94 | 92.06     |
>
> We promise to supplement these discussions together with the new baselines into Section 4 of our paper.

---

> ### Author Response · Authors · 2025-11-27
> **General Response to Common Questions (Part 1 of 3)--Supplementary Note 2**
>
> **Common Question 1.4: Comparison between our agent-based AutoJudger framework and prototype baselines.**
>
> **Response:**
>
> For a comprehensive comparison with our agent-based instantiation of AutoJudger, we choose to conduct experiments on MMT-Bench_VAL, because **MMT-Bench** provides **taxonomies of question categories**, so that we can implement and compare with various baselines, including:
>
> - 4 original baselines (that are widely used in the context of efficient benchmarking of LLMs) that we have previously included in our paper:
>   - **Random**: This method uses pure random selection, no ability decomposition, reporting accuracy as the model ability.
>   - **Stratified**: This method uses weighted random selection, where the selection probability of a question is set as the empirical proportion of its category in the dataset. Reporting accuracy as the model ability.
>   - **Cluster**: This method clusters the questions using BERT embeddings, and selects questions closest to the random sampled cluster centroid. Reporting accuracy as the model ability.
>   - **IRT**: This method uses no ability decomposition, selecting questions with proper difficulty that match the IRT-estimated model ability.
> - 5 extra baselines that we supplement here to demonstrate the extendability of our framework:
>   - **Random + Taxonomy**: This method divides the questions into groups based on categories, and random selects questions within each group. Reporting accuracy as the model ability.
>   - **Random + Taxonomy + IRT**: This method divides the questions into groups based on categories, and prioritize the category whose ability has been least thoroughly probed so far (visited less and with lower average question difficulty), and then randomly choose the next question from that category based on the current IRT estimate of model ability.
>   - **Stratified + IRT**: This method uses weighted random selection, where the selection probability of a question is set as the empirical proportion of its category in the dataset. Using IRT to estimate the model ability.
>   - **IRT + Taxonomy**: This method divides the questions into groups based on the question categories, and within each group, select the questions that matches the IRT-estimated model ability most.
>   - **Hybrid** ($\theta$): To include a setting that selects questions considering both semantics and difficulty of questions, here we propose this hybrid strategy. This method clusters the questions, and selects questions by computing a linear combination of (a) the semantic distance with weight $\theta$ to ensure diversity, and (b) the mismatch between the question difficulty and IRT-estimated model ability with weight 1 to ensure appropriateness. Note that **this strategy is a rigorous controlled version of "AutoJudger w/o agent" variant**, as we have discussed before.
>
> We compare these baselines with our **agent-based instantiation of AutoJudger** (agent-based ability semantic decomposition, semantic and difficulty-aware question selection; IRT-conditioned ability estimation)**.** Results in Rebuttal Table 1 demonstrate that:
>
> - **The design of AutoJudger is flexible.** There are multiple combinations with ability decomposition and estimation, and most cases, when combined with IRT, yield good performance.
> - **The question selection startegy matters.** Random selection consistenly induces inferior performance. The more extra information included (i.e. difficulty and semantics), the better the performance will be. (comparing Exp 7 with Exp 1, Exp 8 with Exp 4, Exp 9-11 with Exp 6)
> - **The agent-based semantic decomposition is effective**, which consistently performs the best across all data budgets, validating our choices of the agent-based instantiation of AutoJudger.
>
> Overall, these results confirm that **our fundamental design choices,** rather than the complexity of the agent itself, **are the main contributors to the observed performance improvements.**

---

> ### Author Response · Authors · 2025-11-27
> **General Response to Common Questions (Part 1 of 3)--Supplementary Note 3**
>
> ***Rebuttal Table 1: Comparison of various methods on MMT-Bench_VAL under varing data budget (1%-5%).*** Here we report the average ranking accuracy by running each method 5 times.
>
> | Exp ID | Method                  | Ability Decomposition     | Ability Estimation | Question Selection    | Data Budget |           |           |           |           |
> | ------ | ----------------------- | ------------------------- | ------------------ | --------------------- | ----------- | --------- | --------- | --------- | --------- |
> |        |                         |                           |                    |                       | **1%**      | **2%**    | **3%**    | **4%**    | **5%**    |
> | 1      | Random                  | /                         | Accuracy           | Random                | 71.32       | 77.94     | 84.26     | 85.15     | 85.88     |
> | 2      | Random + Taxonomy       | taxonomy                  | Accuracy           | Random                | 70.88       | 80.15     | 84.41     | 83.53     | 85.29     |
> | 3      | Random + Taxonomy + IRT | taxonomy                  | IRT                | Random                | 72.65       | 78.82     | 83.53     | 84.85     | 85.00     |
> | 4      | Stratified              | taxonomy                  | Accuracy           | Random                | 70.59       | 76.18     | 85.44     | 84.71     | 82.35     |
> | 5      | Stratified + IRT        | taxonomy                  | IRT                | Random                | 74.26       | 78.68     | 86.47     | 86.32     | 84.12     |
> | 6      | Cluster                 | embedding-based semantics | Accuracy           | Semantic              | 54.12       | 75.88     | 79.85     | 86.62     | 87.79     |
> | 7      | IRT                     | /                         | IRT                | Difficulty            | 76.47       | 80.88     | 82.35     | 82.35     | 88.24     |
> | 8      | IRT + Taxonomy          | taxonomy                  | IRT                | Difficulty            | 81.62       | 83.09     | 85.29     | 90.44     | 90.44     |
> | 9      | Hybrid ($\theta$=0.5)   | embedding-based semantics | IRT                | Difficulty + Semantic | 85.29       | 83.82     | 84.56     | 87.50     | 89.71     |
> | 10     | Hybrid ($\theta$=1)     | embedding-based semantics | IRT                | Difficulty + Semantic | 80.15       | 86.76     | 87.50     | 86.76     | 89.71     |
> | 11     | Hybrid ($\theta$=2)     | embedding-based semantics | IRT                | Difficulty + Semantic | 83.09       | **88.24** | 88.97     | 88.24     | 90.44     |
> | 12     | AutoJudger              | agent-based semantics     | IRT                | Difficulty + Semantic | **85.66**   | 87.87     | **89.34** | **91.91** | **92.10** |

---

### Note · Authors · 2026-01-05

I have read and agree with the venue's withdrawal policy on behalf of myself and my co-authors.